# Distributional LLM-as-a-Judge

**Luyu Chen**[1*]**, Zeyu Zhang**[1*]**, Haoran Tan**[1*]**, Quanyu Dai**[2]**,**
**Hao Yang**[1*]**, Zhenhua Dong**[2]**, Xu Chen**[1†*]
[1]Gaoling School of Artificial Intelligence, Renmin University of China
[2]Huawei Noah's Ark Lab
{luyu.chen,xu.chen}@ruc.edu.cn

## Abstract

LLMs have emerged as powerful evaluators in the LLM-as-a-Judge paradigm, offering significant efficiency and flexibility compared to human judgments. However, previous methods primarily rely on single-point evaluations, overlooking the inherent diversity and uncertainty in human evaluations. This approach leads to information loss and decreases the reliability of evaluations. To address this limitation, we propose a novel training framework that explicitly aligns the LLM-generated judgment distribution with human evaluation distributions. Specifically, we propose a distributional alignment objective based on KL divergence, combined with an auxiliary cross-entropy regularization to stabilize the training process. Furthermore, due to limited human annotations, empirical human distributions are merely noisy estimates of the true underlying distribution. We therefore incorporate adversarial training to ensure a robust alignment with this true distribution, rather than overfitting to its imperfect approximation. Extensive experiments across various LLM backbones and evaluation tasks demonstrate that our framework significantly outperforms existing closed-source LLMs and conventional single-point alignment methods, with superior alignment quality, strong robustness, and competitive evaluation accuracy.

## 1 Introduction

In recent years, large language models (LLMs) have demonstrated remarkable progress across various tasks, such as natural language understanding [1, 2], reasoning [3–5], and evaluation [6, 7]. One of their most significant applications is for automatic judgment, which employs LLMs to evaluate specific targets based on predefined criteria or instructions [8, 9]. This *LLM-as-a-Judge* paradigm offers significant advantages in efficiency and flexibility due to its capability to efficiently handle large-scale data and adapt to diverse evaluation tasks. Therefore, using LLMs as judges has emerged as a promising alternative to conventional human evaluations [10].

Most previous works adopt single-point judgment with LLMs, which just outputs a single result for each sample [11–13]. Although this paradigm is straightforward, it overlooks the inherent diversity of human evaluations. In real-world scenarios, human evaluations are rarely deterministic. Instead, they follow a distribution that encodes valuable signals like the level of consensus and controversy. [14, 15]. Therefore, replacing this distributional human evaluations with a single-point LLM judgment may cause information loss [15], which limits the comprehensiveness and reliability of evaluations. This limitation is particularly critical in high-stakes domains like medical diagnosis or policy-making, where reliance on a single prediction is inherently risky and unreliable [16].

In order to empower LLM judgment with the diversity and uncertainty of human evaluations, an intuitive approach is to generate a judgment distribution based on LLMs. Although LLMs can

---

[†] Corresponding author.
[*] Beijing Key Laboratory of Research on Large Models and Intelligent Governance,
  Engineering Research Center of Next-Generation Intelligent Search and Recommendation, MOE.

39th Conference on Neural Information Processing Systems (NeurIPS 2025).

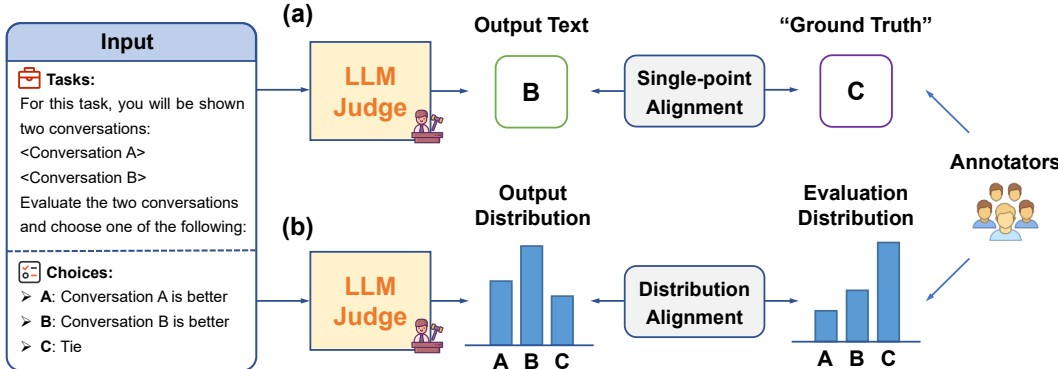

Figure 1: Comparison between single-point alignment and distribution alignment. (a) Single-point Alignment: In this method, LLMs are trained to generate outputs that exactly match the desired text. (b) Distribution Alignment: By using this approach, the models are trained to produce judgment distributions that align with the human evaluation distributions.

inherently provide probability distributions over output tokens, previous studies have shown that they are often overconfident and skewed towards a few options [17]. Besides, most current LLM training approaches focus on single-point alignment, aiming to maximize the probability of generating a specific correct or desired output [18, 19], as illustrated in **Figure 1(a)**. This focus inherently limits their ability to capture the diversity and uncertainty present in human evaluations, hindering effective distribution alignment. Therefore, it is necessary to design an explicit distributional alignment framework to align LLMs' output with human evaluation distributions, as illustrated in **Figure 1(b)**.

To address the above challenges, we design a novel framework that explicitly aligns the output distributions of LLMs with human evaluation distributions. Specifically, we propose a distributional alignment objective that leverages the Kullback–Leibler (KL) divergence [20] to minimize the discrepancy between the model's predicted distribution and the empirical distribution derived from human annotations. Besides, we introduce a hybrid loss function that combines the primary KL divergence objective with an auxiliary cross-entropy loss to improve the training stability. It combines the distributional advantages of KL divergence and the stability of single-point alignment. To further mitigate the risk of overfitting caused by limited human annotations, we propose an adversarial training strategy to improve model robustness. Specifically, we apply the worst-case perturbation to the empirical distributions during optimization, encouraging the model to align with any plausible distribution within the bounded perturbation set. Our major contributions are presented as follows:

• **Explicit Distribution Alignment Framework.** We propose a novel framework to explicitly align the distribution of LLM judgment with human evaluation distributions, thereby effectively capturing the uncertainty and diversity inherent in human evaluations.
• **Robust Distribution Alignment Methodology.** By introducing an adversarial optimization strategy that leverages distribution perturbations during training, we significantly enhance the fidelity and robustness of model alignment with real human evaluation distributions.
• **Extensive Experimental Validations.** Experiments across diverse LLM backbones and evaluation tasks demonstrate that our approach consistently surpasses existing closed-source LLMs and substantially outperforms conventional single-point alignment methods in multiple aspects.

## 2 Related Work

### 2.1 LLM-as-a-Judge

LLMs are increasingly used as automated evaluators (*i.e.,* LLM-as-a-Judge) [11–13, 21, 22] due to their efficiency, scalability, and generalization capabilities [8, 9]. Previous works typically utilize LLMs to produce a single-point deterministic evaluation, such as binary consistency judgments [21] and Likert-scale ratings [22]. However, these approaches neglect the inherent variability observed in human evaluations, where humans often present diverse opinions, resulting in evaluation distributions [14]. Collapsing this diversity into a single decision overlooks valuable information [15] such as disagreement, uncertainty, and subjectivity. To address this limitation, we generate probability distributions from LLMs for evaluations. We propose an explicit alignment method to better match the distributions generated by LLMs with the actual distributions provided by human annotators.

Prior work, such as [23] for the NLI task, has also advocated for learning from full human judgment distributions instead of single labels. Our approach is distinguished by two primary innovations. First, we introduce a novel adversarial training mechanism on the label distribution itself. This mechanism is designed to mitigate the annotation noise stemming from limited data, a known limitation that prior work [23] had not mechanistically solved. Second, we validate our framework's effectiveness in the contemporary LLM-as-a-Judge paradigm, extending its application to modern evaluation tasks like quality evaluation and preference understanding.

## 2.2 Distributional Reward Models

To model diverse human preferences, distributional reward models in Reinforcement Learning from Human Feedback (RLHF) [19] aim to output a distribution over reward values rather than a single scalar reward [24–26]. Existing research in this area typically employs methods that model preference score distributions using mean and variance [24, 25], or adopting quantile regression techniques to achieve finer-grained preference modeling [26]. These approaches often infer reward distributions from human preferences indirectly and necessitate architectural modifications that may decrease the general capabilities of LLMs [27]. Different from previous studies, our proposed approach directly leverages the explicit distributions derived from human evaluations, while it can also preserve the inherent language generation capabilities of LLMs without architectural changes.

## 2.3 Adversarial Training

Adversarial training can enhance model robustness by exposing models to worst-case perturbations in training phase, which has been widely adopted in various fields, such as computer vision [28, 29] and natural language processing [30, 31]. It is often formulated as a min-max optimization problem with two adversarial stages. Specifically, the *maximization* stage identifies the worst-case perturbation, and the *minimization* stage updates the model parameters to minimize loss under these perturbations [32]. This iterative procedure enables the model to learn more robust and reliable decision boundaries. Common optimization algorithms employed in adversarial training include the Fast Gradient Sign Method (FGSM) [33] and Projected Gradient Descent (PGD) [34], both of which have shown strong stability and effectiveness. In this study, we adopt adversarial training to enhance the robustness of LLMs, in order to better align model predictions with human evaluation distributions.

## 3 Preliminary

Consider a dataset $D$, where each sample $x \in D$ is annotated independently by $N$ human annotators. Each annotator assigns labels from a discrete set of categories $\mathcal{C} = \{1, 2, \ldots, C\}$. We define the empirical distribution of human judgments (*i.e.,* human evaluation distribution) for a given sample $x$ as the vector $\mathbf{p}(x) \in \mathbb{R}^C$, whose $i$-th component is given by:

$$p_i(x) = \frac{1}{N} \sum_{j=1}^{N} \mathbb{I}(y_j = i), \quad \forall i \in \mathcal{C}, \tag{1}$$

where $y_j$ denotes the label from the $j$-th annotator for sample $x$, and $\mathbb{I}(\cdot)$ is the indicator function.

Correspondingly, let $\theta$ denote the parameters of the LLM. For an input sample $x$, the model outputs a normalized probability distribution over the $C$ categories. This distribution is represented by the vector $\mathbf{q}_\theta(x)$, where each component $q_{\theta,i}(x)$ indicates the predicted probability for category $i$. Formally, the probability vector $\mathbf{q}_\theta(x)$ is defined as:

$$\mathbf{q}_\theta(x) \in \{\mathbf{q} \in \mathbb{R}^C \mid q_i \geq 0, \ \forall i \in \mathcal{C}, \ \sum_{i=1}^{C} q_i = 1\}. \tag{2}$$

In classical evaluation settings [35, 36], a single deterministic reference label $\mathbf{r}(x) \in \{0, 1\}^C$ is often used, typically defined by selecting the most frequent human annotation:

$$r_i(x) = \begin{cases} 1, & \text{if } i = \arg\max_k p_k(x), \\ 0, & \text{otherwise.} \end{cases} \tag{3}$$

The primary objective of our method is to optimize the model parameters $\theta$ so that the predicted judgement distribution $\mathbf{q}_\theta(x)$ closely aligns with the human judgment distribution $\mathbf{p}(x)$. Our proposed

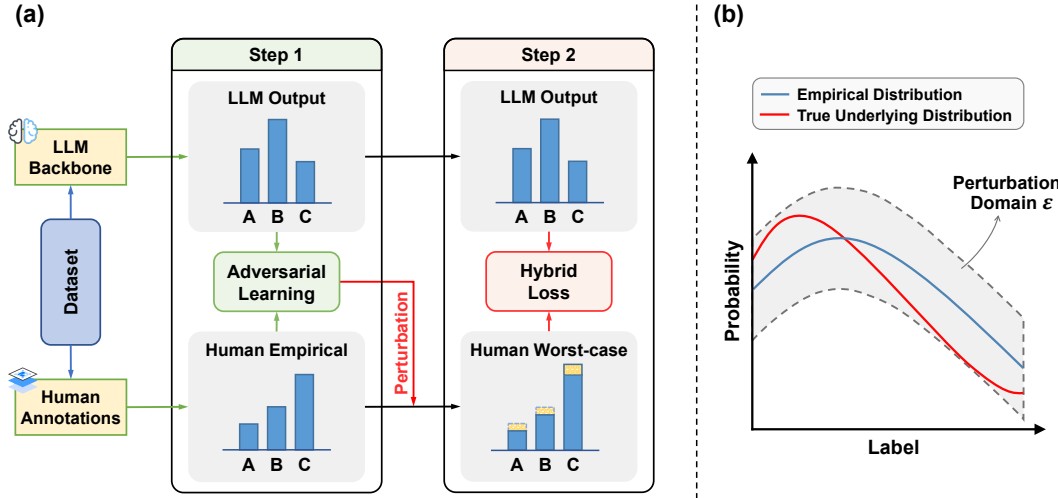

Figure 2: Overview of our proposed framework. (a) Training framework: We generate adversarial perturbations of the empirical human distribution and optimize the hybrid loss. (b) Motivation: Illustrates the relationship between the empirical, perturbed, and true underlying distributions. Robust alignment mitigates the deviation problem in the empirical human distribution.

explicit distributional alignment framework enables the model to better capture nuanced human judgments, thereby resulting in more informative and representative evaluations.

## 4 Methodology

### 4.1 Overview

To overcome the limitations of single-point judgments and better reflect the inherent diversity and uncertainty in human evaluations, we propose a novel training framework that explicitly aligns model-generated probability distributions with real-world human evaluation distributions. Given an input, we extract logits corresponding to the judgment token to obtain the predicted distribution. Our training involves two main steps, as demonstrated in **Figure 2(a)**. First of all, we generate a worst-case perturbation around the empirical distribution to enhance robustness. Then, we compute the hybrid loss between the model prediction and the perturbed distribution, and update the model parameters accordingly. This approach mitigates the inherent limitation of empirical human distributions, which serve only as imperfect estimates of the true underlying distributions, as illustrated in **Figure 2(b)**. By aligning model predictions with all plausible distributions within the perturbation set, our method promotes more robust and faithful distributional alignment.

### 4.2 Human Distribution Alignment via Hybrid Loss

To achieve effective alignment between the model's output distribution and the human judgment distribution, we propose a hybrid loss function. This loss function combines KL divergence for distribution alignment with an auxiliary cross-entropy objective for training stability. First, to explicitly encourage distributional alignment, we introduce the KL divergence loss:

$$\mathcal{L}_{\mathrm{KL}}(\theta) = \frac{1}{|D|} \sum_{x \in D} D_{\mathrm{KL}} \left( \mathbf{p}(x) \parallel \mathbf{q}_\theta(x) \right), \tag{4}$$

where $D_{\mathrm{KL}}(\cdot \parallel \cdot)$ denotes the KL divergence. This objective promotes a fine-grained alignment between the model's predicted distribution and the human evaluation distribution. Second, to improve training stability and guide learning with more direct supervision, we also include the cross-entropy loss as an auxiliary regularizer:

$$\mathcal{L}_{\mathrm{CE}}(\theta) = \frac{1}{|D|} \sum_{x \in D} \mathrm{CE}(\mathbf{q}_\theta(x), \mathbf{r}(x)), \tag{5}$$

where $\mathbf{r}(x)$ is the reference label for sample $x$, determined by majority-vote among the human annotators. This hybrid design mirrors the philosophy of knowledge distillation [37], where student models are trained using both soft targets (through KL divergence from a teacher model) and hard labels (via cross-entropy with ground truth). This approach leverages the rich informational content

provided by the teacher's outputs and the direct guidance of true labels, improving both learning fidelity and convergence stability. Embracing this principle, our hybrid loss function blends these two objectives via a weighting factor $\alpha \in [0, 1]$:

$$\mathcal{L}_{\text{Hybrid}}(\theta) = \alpha \cdot \mathcal{L}_{\text{KL}}(\theta) + (1 - \alpha) \cdot \mathcal{L}_{\text{CE}}(\theta). \tag{6}$$

This hybrid approach ensures stable training using cross-entropy while also achieving nuanced distributional alignment through KL divergence. As a result, it effectively captures both consensus and diversity in human annotations.

### 4.3 Robust Alignment via Adversarial Training

In practice, due to the limited number of human annotations, we only have empirical approximations of the human judgment distribution, as illustrated in **Figure 2(b)**. Directly aligning the model output $\mathbf{q}_\theta(x)$ with these empirical approximations $\mathbf{p}(x)$ results in the model overfitting to sampling noise or artifacts, reducing the robustness of alignment with the true underlying distribution.

To address this challenge, we introduce adversarial training into our distribution alignment framework. Specifically, we define a perturbation set $\mathcal{E}$ around the empirical annotation distribution $\mathbf{p}(x)$ and identify the worst-case perturbed distribution $\mathbf{p}'(x)$ within this set. Aligning our model with this worst-case distribution ensures robustness against any plausible perturbation within $\mathcal{E}$, thereby improving alignment with the human judgment distribution. This transformation converts our objective into a min-max optimization problem as follows:

$$\theta^* = \arg\min_\theta \max_{\mathbf{p}'(x) \in \mathcal{E}} \left[ \alpha \cdot D_{\text{KL}}(\mathbf{p}'(x) \parallel \mathbf{q}_\theta(x)) + (1 - \alpha) \cdot \text{CE}(\mathbf{q}_\theta(x), \mathbf{r}(x)) \right]. \tag{7}$$

This min-max formulation is structurally similar to adversarial training methods like TRADES [38]. However, the two frameworks are conceptually distinct in several key aspects. TRADES perturbs model inputs for attack robustness, with the KL term serving as an auxiliary regularizer for output smoothness. In contrast, we perturb the target label distribution, making KL divergence the primary objective for achieving a more robust and faithful alignment with human judgments.

This adversarial training process consists of two alternating steps:

**1. Adversarial Distribution Generation:** For a fixed model parameter $\theta$ and each sample $x$ in the batch, find the worst-case distribution $\mathbf{p}'(x)$ within the perturbation set $\mathcal{E}$ that maximizes the loss.
**2. Model Update:** Update model parameters $\theta$ to minimize the adversarially perturbed loss.

Through this adversarial training procedure, we explicitly model the worst-case scenarios of the true underlying human judgment distribution, thereby ensuring robust and stable alignment. By accounting for potential annotation noise and sampling artifacts, the model becomes less sensitive to empirical inaccuracies, thus enhancing its generalization performance and practical applicability.

### 4.4 Implementing Adversarial Training via Projected Gradient Descent

To solve the inner maximization equation in Equation (7), we adopt Projected Gradient Descent (PGD) to iteratively search for the worst-case perturbation within a constrained space.. Specifically, we seek an adversarial distribution $\mathbf{p}'(x)$ that maximizes the KL divergence from the model prediction $\mathbf{q}_\theta(x)$, while remaining close to the original human distribution $\mathbf{p}(x)$ and preserving the properties of a valid probability distribution.

We define the feasible perturbation set $\mathcal{E}$ as the intersection of two convex sets as $\mathcal{E} = \Delta^C \cap \mathcal{B}_{\epsilon^*}(\mathbf{p}(x))$, where $\Delta^C = \{\mathbf{p}' \in \mathbb{R}^C \mid \sum_{i=1}^C p_i' = 1, \ p_i' \geq 0\}$ is the $C$-dimensional probability simplex, and $\mathcal{B}_{\epsilon^*}(\mathbf{p}(x))$ is an $\ell_2$ ball of radius $\epsilon^*$ centered at the original human distribution $\mathbf{p}(x)$. Based on the feasible perturbation set, we design the optimization procedure as follows. First of all, we initialize the unperturbed distribution as $\mathbf{p}^{(0)} = \mathbf{p}(x)$. Then, we conduct the gradient ascent by updating the current iterate in the direction of the gradient of the KL divergence as:

$$\mathbf{y}^{(t+1)} = \mathbf{p}^{(t)} + \eta \cdot \nabla_{\mathbf{p}^{(t)}} \left[ D_{\text{KL}}(\mathbf{p}^{(t)} \parallel \mathbf{q}_\theta(x)) \right], \tag{8}$$

where $\eta$ denotes the step size controlling the gradient ascent magnitude. After that, we project the updated distribution back onto the feasible set $\mathcal{E}$ with the equation:

$$\mathbf{p}^{(t+1)} = \Pi_\mathcal{E}(\mathbf{y}^{(t+1)}) = \arg\min_{\mathbf{p}' \in \mathcal{E}} \|\mathbf{p}' - \mathbf{y}^{(t+1)}\|_2^2. \tag{9}$$

This projection step is a convex Quadratically Constrained Quadratic Program (QCQP) [39], as it minimizes a convex quadratic objective over the intersection of two convex sets: the simplex and the $\ell_2$ ball. Notably, the intersection $\mathcal{E}$ is guaranteed to be non-empty since the original distribution $\mathbf{p}(x) \in \mathcal{E}$ by definition. Consequently, this projection problem is well-posed and can be efficiently solved using off-the-shelf convex optimization solvers such as CVXPY [40].

## 5 Experiments

### 5.1 Experiment Setup

**Datasets.** We evaluate our framework using representative datasets [15] from three fundamental LLM-as-a-Judge applications: dataset labeling, quality evaluation, and pairwise preference prediction.

- **Dataset Labeling (SNLI [41]/MNLI [42]).** We use the classic NLI benchmarks to represent the dataset labeling task, where the goal is to determine the logical relationship (entailment, neutral, contradiction) between two sentences. Each instance is annotated by five distinct raters, providing the necessary label distribution. We randomly sample an equal number of instances from MNLI (10,000 each) to maintain a comparable data scale with SNLI.

- **Quality Evaluation (SummEval [35]).** To evaluate performance on text quality assessment, we use the SummEval dataset. This benchmark contains machine-generated summaries of news articles from the CNN/DailyMail corpus. For each summary, quality ratings are provided by a group of experts and crowdworkers on a 1-5 Likert scale across four dimensions (fluency, coherence, consistency, and relevance), forming a rich distributional signal of perceived quality. We treat each dimension as an independent evaluation instance.

- **Pairwise Preference Prediction (MT-Bench [43]).** For the task of understanding human preferences, we use the MT-Bench dataset. For each dialogue, preferences between two model responses (A vs. B) are collected from multiple human reviewers. This yields a preference distribution (A is better, B is better, or Tie) for each comparison, directly reflecting the consensus and disagreement in human choices.

All datasets are split into training and test sets at an 8:2 ratio in our experiments. To facilitate the reproduction, we present the detailed prompts that are used in all the tasks in **Appendix F**.

**Baselines.** We compare our proposed approach against two baseline methods. **(1) Raw Model:** We directly evaluate the pretrained LLMs without any task-specific fine-tuning. This baseline aims to measure the inherent alignment between pretrained models and human judgment distributions, reflecting the model's original capability to approximate human judgment without explicit training or adjustment. **(2) Single-point Alignment:** We adopt the traditional supervised fine-tuning strategy [19], using only the most frequent human annotation label as the supervision target. This baseline evaluates the effectiveness of conventional single-point alignment methods.

**Models.** Our study evaluates both open-source (Qwen2.5-7B [44], LLaMA3.1-8B [45]) and closed-source (GPT-4o, GPT-4o-mini) language models. Closed-source models, recognized for their strong performance, are utilized without additional tuning, whereas the chosen open-source models are tested both with and without further training.

**Training Details.** We fine-tune all selected open-source models using Low-Rank Adaptation (LoRA) [46] with a uniform hyperparameter configuration to ensure a fair comparison. Specifically, we employ the AdamW [47] optimizer with a learning rate of $5 \times 10^{-5}$ and train each model for 2 epochs. To enhance model robustness, we incorporate adversarial training, setting the perturbation step size to 0.05 and performing 5 gradient ascent steps per training iteration. Additionally, we conduct a hyperparameter search for two critical parameters: the weight parameter $\alpha$, chosen from the set $\{0, 0.2, 0.4, 0.6, 0.8, 1.0\}$, and the perturbation radius parameter $\epsilon$, selected from the set $\{0.0, 0.05, 0.1, 0.15, 0.2, 0.25\}$. We conduct all experiments on one NVIDIA A100-40G GPU.

**Evaluation Metrics.** We employ two metrics to measure the alignment between model-predicted and human-annotated distributions. KL Divergence is our primary metric, while Accuracy serves as a complementary measure:
(1) KL Divergence: As our primary measure of success, this metric directly quantifies the discrepancy between the model's predicted distribution $\mathbf{q}_\theta(x)$ and the human distribution $\mathbf{p}(x)$. Lower KL

Table 1: Main results comparing raw models, single-point alignment, and our distribution alignment method across four datasets. KL indicates KL divergence, and Acc denotes top-1 accuracy. Results for fine-tuned models (Single-point and Distribution) are averaged over 5 runs. The * indicates a statistically significant improvement over the single-point baseline ($p < 0.05$).

| Model | Method | SNLI | | MNLI | | Summeval | | MT-Bench | |
|---|---|---|---|---|---|---|---|---|---|
| | | KL↓ | Acc↑ | KL↓ | Acc↑ | KL↓ | Acc↑ | KL↓ | Acc↑ |
| GPT-4o-mini | Raw model | 2.13 | 87.0% | 1.88 | 84.9% | 5.23 | 25.6% | 5.63 | 62.3% |
| GPT-4o | Raw model | 1.75 | 85.5% | 1.16 | 84.2% | 2.82 | 35.2% | 2.48 | 68.5% |
| Qwen2.5 | Raw model | 2.08 | 83.1% | 1.77 | 83.5% | 4.94 | 22.7% | 3.36 | 62.0% |
| | Single-point | 0.60 | 92.7% | 0.64 | 89.7% | 0.73 | 45.6% | 0.82 | 64.0% |
| | Distribution (Ours) | **0.23**\* | **93.3%**\* | **0.23**\* | 89.8% | **0.53**\* | 45.9% | **0.68**\* | **65.4%** |
| LLaMA3.1 | Raw model | 0.90 | 64.9% | 0.67 | 70.5% | 3.60 | 29.5% | 1.58 | 53.4% |
| | Single-point | 0.69 | 92.4% | 0.67 | 89.6% | 0.67 | 45.7% | 0.81 | 62.1% |
| | Distribution (Ours) | **0.28**\* | 92.4% | **0.24**\* | **90.0%**\* | **0.51**\* | **47.3%**\* | **0.74**\* | 62.8% |

divergence indicates closer alignment:

$$\mathbf{KL}(\mathbf{p}(x)||\mathbf{q}_\theta(x)) = \sum_i \mathbf{p}(x) \log \frac{\mathbf{p}(x)}{\mathbf{q}_\theta(x)}. \tag{10}$$

(2) Accuracy: We include Accuracy as a secondary metric for two practical reasons. First, it serves as a valuable indicator of our model's ability to capture the majority consensus in human judgments. More critically, it provides a bridge for comparison with prior work that relies solely on this traditional standard. It measures whether the model's most probable predicted label aligns with the most frequent label in the human judgment distribution:

$$\text{Accuracy} = \frac{1}{|D|} \sum_{x \in D} \mathbb{I}\left(\arg\max_{i \in \mathcal{C}} q_{\theta,i}(x) = \arg\max_{j \in \mathcal{C}} p_j(x)\right). \tag{11}$$

Here, $|D|$ denotes the total number of test samples, $q_{\theta,i}(x)$ represents the model-predicted probability for category $i$, and $p_j(x)$ denotes the human judgment distribution for category $j$.

**Extracting Model Predictions.** We extract model predictions by retrieving logits corresponding to potential judgment labels (e.g., "entailment", "neutral", "contradiction" for NLI tasks, or 1-5 for Likert-scale ratings). These logits are converted into probabilities via softmax normalization. To handle variations in tokenization, the probabilities of synonymous tokens are aggregated into a standard label. For example, the probabilities for tokens like "contra" or "contradict" are summed and assigned to the canonical "contradiction" label. To maintain consistency across experiments, we limit the extraction to the top-5 logits because OpenAI restricts the number of logits returned. Besides, the logits beyond the fifth highest are generally negligible, with values often falling below 1e-6.

## 5.2 Overall Performance

We evaluate the effectiveness of our distribution alignment method across four benchmark datasets, including SNLI, MNLI, Summeval, and MT-Bench. The primary experimental results are summarized in **Table 1**. Detailed results for all fine-tuned models, including mean and standard deviation, are provided in Appendix E. Our findings highlight three key aspects as follows:

**(1) Necessity of Distribution Alignment.** Without specific alignment training, both open-source and closed-source models demonstrate substantial divergence between their predictions and human judgment, with KL divergence typically exceeding 2.0. This indicates that current models inherently produce judgment distributions that are misaligned with human evaluations, highlighting the necessity for additional training of distribution alignment.

**(2) Superiority of Our Proposed Method.** Compared to conventional single-point alignment methods, our approach consistently achieves better distribution alignment across different datasets and LLM backbones. It significantly reduces KL divergence while maintaining accuracy, demonstrating its generalization ability and effectiveness in aligning model outputs with human-labeled distributions.

Table 2: Ablation study of our proposed method. We analyze the contribution of adversarial training (Adv), KL divergence loss (KL), and cross-entropy loss (CE) on MNLI and Summeval datasets.

| Method | Components | | | Qwen2.5 | | | | LLaMA3.1 | | | |
| | Adv | KL | CE | MNLI | | Summeval | | MNLI | | Summeval | |
| | | | | KL↓ | Acc↑ | KL↓ | Acc↑ | KL↓ | Acc↑ | KL↓ | Acc↑ |
|---|---|---|---|---|---|---|---|---|---|---|---|
| Raw Model | - | - | - | 1.77 | 83.5% | 4.94 | 22.7% | 0.67 | 70.5% | 3.60 | 29.5% |
| Single-point | - | - | ✓ | 0.64 | 89.7% | 0.73 | 45.6% | 0.67 | 89.6% | 0.67 | 45.7% |
| Ours (Full) | ✓ | ✓ | ✓ | **0.23** | **89.8%** | **0.53** | 45.9% | **0.24** | **90.0%** | **0.51** | 47.3% |
| Ours w/o Adv | - | ✓ | ✓ | 0.25 | 89.0% | 0.64 | 46.6% | 0.32 | 89.6% | 0.62 | 45.9% |
| Ours w/o KL | ✓ | - | ✓ | 0.78 | 88.4% | 0.75 | 45.8% | 0.65 | 89.2% | 0.65 | 45.4% |
| Ours w/o CE | ✓ | ✓ | - | 0.23 | 89.0% | 0.54 | 46.0% | 0.33 | 88.7% | 0.58 | 48.0% |

**(3) Correlation between Model Capability and Alignment Performance.** We observe a positive correlation between a model's inherent capability and its alignment performance. More capable models, such as GPT-4o, not only achieve higher accuracy but also yield predicted distributions closer to human annotations than weaker models like GPT-4o-mini and Qwen2.5. This suggests that stronger models naturally produce judgments distributions more consistent with human evaluations.

In conclusion, our method demonstrates superior performance in distribution alignment compared to raw models and conventional single-point alignment approaches. It reduces KL divergence while maintaining accuracy across multiple datasets and various LLM backbones, presenting the effectiveness of our method in aligning model outputs with human judgment distributions.

## 5.3 Ablation Study

To better understand the contribution of each component in our method, we conduct an ablation study, whose results are summarized in **Table 2**. We observe that removing any single component consistently degrades alignment performance, resulting in increased KL divergence. This indicates that all three components complement each other and collectively enhance distributional alignment.

According to the results, we find that KL divergence loss is the most crucial. Removing it leads to a significant increase in KL divergence, which confirms its essential role in human judgment alignment by penalizing deviations from the target distribution. Besides, adversarial training also contributes to improving the performance. By introducing perturbations during training, the model can align better with human distributions even under worst-case distributional shifts, thereby improving robustness and generalization. In addition, removing cross-entropy loss results in only a slight increase in KL divergence. Although its effect on distributional alignment is limited, extensive experiments in **Section** 5.4 demonstrate that a small amount of auxiliary CE loss can stabilize training.

## 5.4 Impact of Hyper-parameters

We further perform experiments on how the weighting parameter $\alpha$ and perturbation radius $\epsilon$ affect model alignment performance. Using Qwen2.5 as the base model, we evaluate its alignment performance across all four datasets. Lower KL divergence indicates better alignment between the model prediction and the human judgment distribution. The results are presented in **Figure 3**.

**Impact of Weighting Parameter $\alpha$.** The weighting parameter $\alpha$ balances the KL divergence and CE losses, thereby affecting the performance of alignment. We observe that increasing $\alpha$ from 0 to approximately 0.8 consistently enhances the alignment performance across all datasets. However, removing the CE term entirely ($\alpha$=1.0) leads to a noticeable performance decline. This suggests that a small portion of the CE loss is crucial for stabilizing the training process. This instability is especially acute on MT-Bench. We hypothesize this stems from the task's high difficulty and subjectivity, which creates a more complex target distribution. For such challenging distributions, a pure KL divergence objective can become unstable.

**Impact of Perturbation Radius $\epsilon$.** We further explore the influence of the perturbation radius $\epsilon$ in our method. Each line in **Figure 3** corresponds to a specific value of perturbation radius, and we use the blue line ($\epsilon = 0$) to represent training without adversarial perturbations. The results demonstrate that

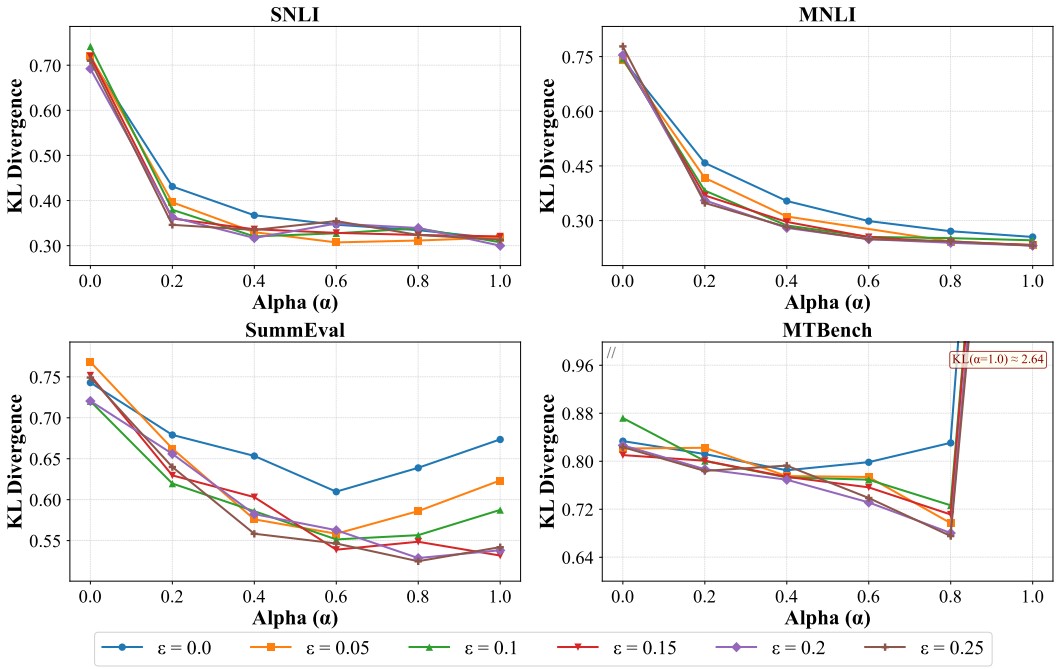

Figure 3: Effect of weighting parameter $\alpha$ and perturbation radius $\epsilon$ on KL divergence across four datasets. Lower values indicate better alignment between model predictions and human distributions.

the method without adversarial training commonly performs the worst. It indicates that adversarial training can enhance the model's generalization capability, thereby improving alignment performance. As the perturbation parameter $\epsilon$ increases, the alignment performance exhibits a general improvement. However, the performance gains gradually diminish with increasing perturbation magnitudes, and this trend is particularly evident on the MNLI dataset.

These results have verified our earlier statements: KL divergence serves as the primary mechanism for alignment, while incorporating a minor component of cross-entropy loss can further improve the training stability. It further underscores the importance of combining both components. Besides, the integration of moderate adversarial perturbations further boosts alignment performance by increasing the model's robustness against the shifts in real-world human evaluation distributions.

### 5.5 Robustness Analysis

To further analyze the robustness of our method, we conduct extensive experiments by adding random perturbations to the target label distributions in the test set. Specifically, our random perturbations $\delta$ are ranged from 0.00 to 0.25. The experiments are performed on all four datasets based on Qwen2.5, and we use the hyper-parameters identified in **Section 5.4** with $\alpha = 0.8$ and $\epsilon = 0.25$.

As shown in **Table 3**, our full method consistently achieves the lowest KL divergence across all perturbation levels and datasets, outperforming both the single-point alignment baseline and the variant without adversarial training. These results suggest that incorporating adversarial training enables the model to effectively align with all plausible distributions within the perturbation set, thereby improving robustness and fidelity in distributional alignment.

## 6 Conclusion

In this paper, we propose a distribution alignment framework that explicitly aligns the model outputs with the human evaluation distribution, aiming for more nuanced evaluation results. Specifically, we employ KL divergence as the main objective to minimize the discrepancy between model predictions and target distributions. Furthermore, we introduce a hybrid loss function, incorporating an auxiliary cross-entropy loss to stabilize training. Finally, adversarial training is utilized to further enhance alignment performance by increasing the model's robustness against distributional shifts. Experiments demonstrate that our distribution alignment method outperforms existing single-point alignment approaches and exhibits strong generalization and robustness across different models and datasets.

Table 3: The robustness analysis of our distribution alignment method under varying label perturbation levels ($\delta$). The KL divergences are reported across different datasets and models, with lower KL divergence indicating better performance in distribution alignment.

| Dataset | Method | KL Divergence at different perturbation levels ($\delta$) | | | | | |
|---|---|---|---|---|---|---|---|
| | | $\delta = 0.00$ | $\delta = 0.05$ | $\delta = 0.10$ | $\delta = 0.15$ | $\delta = 0.20$ | $\delta = 0.25$ |
| **SNLI** | Single-point | 0.715 | 0.717 | 0.708 | 0.692 | 0.720 | 0.709 |
| | Ours w/o Adv | 0.334 | 0.336 | 0.333 | 0.327 | 0.344 | 0.346 |
| | Ours (Full) | **0.324** | **0.325** | **0.323** | **0.317** | **0.335** | **0.336** |
| **MNLI** | Single-point | 0.742 | 0.744 | 0.743 | 0.745 | 0.753 | 0.757 |
| | Ours w/o Adv | 0.271 | 0.272 | 0.272 | 0.276 | 0.283 | 0.293 |
| | Ours (Full) | **0.243** | **0.245** | **0.245** | **0.251** | **0.256** | **0.264** |
| **Summeval** | Single-point | 0.743 | 0.748 | 0.752 | 0.778 | 0.809 | 0.836 |
| | Ours w/o Adv | 0.639 | 0.643 | 0.649 | 0.669 | 0.707 | 0.735 |
| | Ours (Full) | **0.525** | **0.529** | **0.539** | **0.565** | **0.588** | **0.612** |
| MT-**Bench** | Single-point | 0.833 | 0.833 | 0.837 | 0.832 | 0.836 | 0.848 |
| | Ours w/o Adv | 0.831 | 0.830 | 0.834 | 0.829 | 0.832 | 0.846 |
| | Ours (Full) | **0.675** | **0.676** | **0.678** | **0.675** | **0.677** | **0.689** |

## Limitations

Although our approach can effectively align model outputs with human judgment distributions, it exhibits two notable limitations. First, the model's explainability is limited, as the generated explanations only correspond to a single sampled judgment rather than interpreting the entire predicted distribution. Second, suitable training datasets remain scarce due to high human annotation costs. Most existing datasets contain only a single annotation per instance. Therefore, improving model alignment across diverse tasks requires constructing more datasets with richer human evaluation data. In future work, we will further improve the explainability and efficiency of our proposed method.

## Acknowledgments and Disclosure of Funding

This work is supported in part by National Natural Science Foundation of China (No. 62422215 and No. 62472427), Major Innovation & Planning Interdisciplinary Platform for the "DoubleFirst Class" Initiative, Renmin University of China, Public Computing Cloud, Renmin University of China, fund for building world-class universities (disciplines) of Renmin University of China, the Outstanding Innovative Talents Cultivation Funded Programs 2024 of Renmin University of China, and Huawei Innovation Research Programs. We gratefully acknowledge the support from Mindspore[1], CANN(Compute Architecture for Neural Networks) and Ascend AI Processor used for this research.

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

# A    Effectiveness under a Fixed Annotation Budget

A critical question for our distributional framework is whether its superior performance stems from its inherent methodology, or if it is merely an artifact of using more human annotations than single-point methods. To answer this question, this section empirically investigates the trade-off between annotating a larger number of unique samples (sample breadth) versus collecting multiple judgments for each sample (annotation depth), while keeping the total number of collected annotations constant.

## A.1    Experimental Setup

We designed a controlled experiment on the SNLI dataset with a fixed budget of approximately 8,000 total human annotations. We compare three distinct data allocation strategies:

- **Strategy 1 (Breadth-Focused):** 8,000 unique samples were used, each paired with **one** randomly selected human annotation. This strategy maximizes sample breadth to represent the conventional single-point alignment approach.

- **Strategy 2 (Balanced):** 2,667 unique samples were used, each paired with **three** randomly selected human annotations. This strategy represents a balanced trade-off between sample breadth and annotation depth.

- **Strategy 3 (Depth-Focused):** 1,600 unique samples were used, each paired with all **five** available human annotations. This strategy maximizes annotation depth over a smaller set of unique samples.

All strategies were trained using our distributional alignment framework. Notably, for Strategy 1, our method's objective simplifies to become equivalent to traditional single-point alignment. All models were trained under identical conditions for fair comparison.

## A.2    Results and Analysis

Table 4: Performance comparison under a fixed annotation budget. The balanced strategy (Strategy 2) achieves the best distributional alignment (KL Divergence) and is time-efficient.

| Annotation Strategy | Time/Epoch (min) | KL Divergence ($\downarrow$) | Accuracy ($\uparrow$) |
|:---:|:---:|:---:|:---:|
| Strategy 1 | 21.9 | $0.32_{\pm 0.01}$ | $89.1\%_{\pm 0.4\%}$ |
| Strategy 2 | 9.6 | $\mathbf{0.25}^*_{\pm 0.00}$ | $88.9\%_{\pm 0.2\%}$ |
| Strategy 3 | 5.7 | $0.29_{\pm 0.00}$ | $89.1\%_{\pm 0.1\%}$ |

The results, summarized in Table 4, highlight two key findings. First, the balanced approach (Strategy 2) achieves the best distributional alignment, yielding a lower KL Divergence than the other strategies. Compared to the breadth-focused strategy (Strategy 1), this underscores the importance of a distributional signal for effective alignment. Compared to the depth-focused strategy (Strategy 3), it suggests that maximizing annotation depth at the expense of sample variety can hurt generalization, likely due to the model overfitting to a smaller set of examples.

Second, the distributional approaches (Strategies 2 and 3) are substantially more computationally efficient. By processing a smaller number of unique samples per epoch, they dramatically reduce training time. This analysis indicates that collecting a moderate number of judgments per sample is a more effective and efficient strategy for distributional alignment.

# B    Further Validation on Modern Benchmarks

To further assess the effectiveness of our framework, we conducted additional experiments on three modern benchmarks. These datasets correspond to three fundamental LLM-as-a-Judge applications but feature more challenging data, including denser annotations, more contemporary model outputs, and greater sample diversity.

## B.1    Benchmark Datasets

- **ChaosNLI** [48] serves as a highly robust benchmark for the **Dataset Labeling** task. In contrast to SNLI's 5 annotations per instance, ChaosNLI was specifically created to study the full spectrum

of human opinion by collecting 100 annotations for each of the 3,113 examples, providing an exceptionally dense and reliable ground-truth distribution.

- **HelpSteer2** [49] provides a modern benchmark for the **Quality Evaluation** of LLM-generated responses. Unlike SummEval, which evaluates outputs from specialized summarization models, HelpSteer2 focuses on rating the outputs of contemporary LLMs across five dimensions: helpfulness, correctness, coherence, complexity, and verbosity. Each response is rated by multiple annotators on a Likert-5 scale, and we follow our main experimental protocol by treating each dimension as an independent evaluation instance.

- **HelpSteer2-Preference** [50] is a fine-grained benchmark for **Pairwise Preference Prediction**. Similar to MT-Bench, it involves human annotators indicating their preference between two LLM responses. However, it offers a more nuanced 6-point rating scale (from -3 to +3) for the degree of preference. To maintain consistency with our experimental setup, we mapped these scores to a 3-point scale (A is better, B is better, or Tie), where scores of {-3, -2} correspond to one preference, scores of {-1, 1} correspond to a tie, and scores of {2, 3} correspond to the other preference.

### B.2 Results and Analysis

The results on these modern benchmarks are presented in Table 5. The findings are highly consistent with the conclusions from our main experiments presented in the paper.

Table 5: Results on modern benchmarks for dataset labeling (ChaosNLI), quality evaluation (Help-Steer2), and preference prediction (HelpSteer2-Preference). Our method consistently outperforms baselines in KL Divergence while achieving competitive or superior accuracy. The * indicates that the improvement of our method over the single-point baseline is statistically significant ($p < 0.05$).

| Model | Method | ChaosNLI | | HelpSteer2 | | HelpSteer2-Pref. | |
|---|---|---|---|---|---|---|---|
| | | KL↓ | Acc↑ | KL↓ | Acc↑ | KL↓ | Acc↑ |
| GPT-4o-mini | Raw model | $3.92_{\pm0.00}$ | $64.1\%_{\pm0.0\%}$ | $4.83_{\pm0.00}$ | $42.4\%_{\pm0.0\%}$ | $13.8_{\pm0.00}$ | $9.2\%_{\pm0.0\%}$ |
| GPT-4o | Raw model | $2.43_{\pm0.00}$ | $61.2\%_{\pm0.0\%}$ | $2.09_{\pm0.00}$ | $40.4\%_{\pm0.0\%}$ | $4.98_{\pm0.00}$ | $18.9\%_{\pm0.0\%}$ |
| | Raw model | $3.94_{\pm0.00}$ | $60.3\%_{\pm0.0\%}$ | $3.79_{\pm0.00}$ | $32.0\%_{\pm0.0\%}$ | $7.65_{\pm0.00}$ | $10.0\%_{\pm0.0\%}$ |
| Qwen2.5-7B | Single-point | $1.22_{\pm0.02}$ | $70.6\%_{\pm0.7\%}$ | $0.76_{\pm0.03}$ | $60.5\%_{\pm0.1\%}$ | $0.57_{\pm0.02}$ | $71.4\%_{\pm0.8\%}$ |
| | Distribution (Ours) | $\mathbf{0.41}^*_{\pm0.02}$ | $\mathbf{71.8\%}^*_{\pm0.5\%}$ | $\mathbf{0.63}^*_{\pm0.01}$ | $60.0\%_{\pm0.5\%}$ | $\mathbf{0.49}^*_{\pm0.01}$ | $71.3\%_{\pm1.1\%}$ |
| | Raw model | $0.68_{\pm0.00}$ | $57.8\%_{\pm0.0\%}$ | $2.50_{\pm0.00}$ | $13.3\%_{\pm0.0\%}$ | $2.86_{\pm0.00}$ | $14.9\%_{\pm0.0\%}$ |
| LLaMA3.1-8B | Single-point | $1.14_{\pm0.04}$ | $65.0\%_{\pm0.5\%}$ | $0.73_{\pm0.02}$ | $62.4\%_{\pm0.3\%}$ | $0.51_{\pm0.01}$ | $71.6\%_{\pm0.5\%}$ |
| | Distribution (Ours) | $\mathbf{0.43}^*_{\pm0.05}$ | $65.7\%_{\pm1.2\%}$ | $\mathbf{0.59}^*_{\pm0.00}$ | $62.4\%_{\pm0.1\%}$ | $\mathbf{0.47}^*_{\pm0.00}$ | $\mathbf{73.8\%}^*_{\pm0.3\%}$ |

Across these benchmarks, our framework consistently outperforms the baselines in KL Divergence, often by a significant margin. At the same time, it maintains competitive or superior accuracy. Consistent with our main experimental findings, these results provide strong additional evidence that our approach is robust, generalizable, and highly effective for modern LLM-as-a-Judge applications.

## C  Out-of-Distribution Generalization

To assess our framework's generalization capabilities, we conducted an out-of-distribution (OOD) experiment. Specifically, models were fine-tuned exclusively on the **SNLI** training set. Subsequently, they were evaluated directly on the unseen **ChaosNLI** test set, without any further training or adaptation. ChaosNLI serves as a suitable OOD target due to its shared task formulation but distinct data source and significantly denser annotation distribution. The results are presented in Table 6.

As shown in the table, even when faced with an unseen dataset, our framework significantly outperforms the single-point alignment baseline in both KL Divergence and Accuracy. This result confirms that our method learns a robust and transferable representation of human disagreement.

## D  Analysis of Computational Efficiency

A key concern with adversarial training is that the inner PGD optimization loop could introduce significant computational overhead. However, in our framework, this overhead is minimal. Our PGD procedure computes gradients with respect to the target label distribution $\mathbf{p}(x)$, not the language

Table 6: OOD experiment results from training on SNLI and evaluating on the unseen ChaosNLI test set. The * indicates a statistically significant improvement over the single-point baseline ($p < 0.05$).

| Model | Method | ChaosNLI (OOD) | |
|---|---|---|---|
| | | KL↓ | Acc↑ |
| Qwen2.5-7B | Raw model | $3.94_{\pm 0.00}$ | $60.3\%_{\pm 0.0\%}$ |
| | Single-point | $1.01_{\pm 0.01}$ | $66.5\%_{\pm 0.9\%}$ |
| | Distribution (Ours) | $\mathbf{0.31}^{*}_{\pm 0.01}$ | $\mathbf{67.8\%}^{*}_{\pm 0.5\%}$ |
| LLaMA3.1-8B | Raw model | $0.68_{\pm 0.00}$ | $57.8\%_{\pm 0.0\%}$ |
| | Single-point | $1.15_{\pm 0.05}$ | $60.3\%_{\pm 0.1\%}$ |
| | Distribution (Ours) | $\mathbf{0.46}^{*}_{\pm 0.02}$ | $\mathbf{62.7\%}^{*}_{\pm 0.1\%}$ |

model's parameters $\theta$. During these inner steps, the model's output $\mathbf{q}_\theta(x)$ is treated as a fixed constant, thus avoiding any costly backpropagation through the language model.

To empirically quantify this overhead, we benchmarked the training efficiency on the MNLI dataset with the Qwen2.5-7B model on a single NVIDIA A100 GPU. As shown in Table 7, our method introduces a modest slowdown of approximately 21% compared to the standard single-point baseline. We contend that this is an acceptable trade-off for the significant improvements in alignment quality and robustness.

Table 7: Training efficiency comparison on the MNLI dataset. Our method incurs a modest overhead for a significant gain in alignment performance.

| Method | Time / Epoch (min) | Throughput (samples/sec) | Relative Slowdown |
|---|---|---|---|
| Single-point | 23.4 | 5.70 | 1.0× |
| Distributional (Ours) | 28.3 | 4.71 | 1.21× |

# E    Detailed Main Results with Standard Deviations

This section provides the detailed experimental results for the open-source models presented in Section 5.2. We report the mean and standard deviation over 5 runs. Table 8 presents the results for the NLI tasks, and Table 9 presents the results for the evaluation tasks.

Table 8: Detailed results (mean ± std over 5 runs) for NLI tasks (SNLI and MNLI). The * indicates a statistically significant improvement over the single-point baseline ($p < 0.05$).

| Model | Method | SNLI | | MNLI | |
|---|---|---|---|---|---|
| | | KL↓ | Acc↑ | KL↓ | Acc↑ |
| Qwen2.5 | Single-point | $0.60_{\pm 0.01}$ | $92.7\%_{\pm 0.1\%}$ | $0.64_{\pm 0.02}$ | $89.7\%_{\pm 0.2\%}$ |
| | Distribution (Ours) | $\mathbf{0.23}^{*}_{\pm 0.01}$ | $\mathbf{93.3\%}^{*}_{\pm 0.2\%}$ | $\mathbf{0.23}^{*}_{\pm 0.00}$ | $89.8\%_{\pm 0.2\%}$ |
| LLaMA3.1 | Single-point | $0.69_{\pm 0.02}$ | $92.4\%_{\pm 0.42\%}$ | $0.67_{\pm 0.02}$ | $89.6\%_{\pm 0.2\%}$ |
| | Distribution (Ours) | $\mathbf{0.28}^{*}_{\pm 0.01}$ | $92.4\%_{\pm 0.13\%}$ | $\mathbf{0.24}^{*}_{\pm 0.02}$ | $\mathbf{90.0\%}^{*}_{\pm 0.2\%}$ |

# F    Prompts

**Summeval.** These prompts are reused from G-Eval[36] with slight modifications. Specifically, the output label region has been explicitly specified, and the model is instructed to directly output evaluation results without providing explanations.

Table 9: Detailed results (mean $\pm$ std over 5 runs) for evaluation tasks (Summeval and MT-Bench). The * indicates a statistically significant improvement over the single-point baseline ($p < 0.05$).

| Model | Method | Summeval | | MT-Bench | |
|---|---|---|---|---|---|
| | | KL↓ | Acc↑ | KL↓ | Acc↑ |
| Qwen2.5 | Single-point | $0.73_{\pm 0.03}$ | $45.6\%_{\pm 0.5\%}$ | $0.82_{\pm 0.01}$ | $64.0\%_{\pm 1.0\%}$ |
| | Distribution (Ours) | $\mathbf{0.53}^*_{\pm 0.02}$ | $45.9\%_{\pm 0.4\%}$ | $\mathbf{0.68}^*_{\pm 0.02}$ | $\mathbf{65.4\%}_{\pm 1.4\%}$ |
| LLaMA3.1 | Single-point | $0.67_{\pm 0.05}$ | $45.7\%_{\pm 0.8\%}$ | $0.81_{\pm 0.01}$ | $62.1\%_{\pm 0.8\%}$ |
| | Distribution (Ours) | $\mathbf{0.51}^*_{\pm 0.01}$ | $\mathbf{47.3\%}^*_{\pm 0.6\%}$ | $\mathbf{0.74}^*_{\pm 0.01}$ | $62.8\%_{\pm 0.8\%}$ |

---

**Prompts Used for Summeval for Coherence Evaluation**

You will be given one summary written for a news article.

Your task is to rate the summary on one metric.

Please make sure you read and understand these instructions carefully. Please keep this document open while reviewing, and refer to it as needed.

Evaluation Criteria:

Coherence (1-5) - the collective quality of all sentences. We align this dimension with the DUC quality question of structure and coherence whereby "the summary should be well-structured and well-organized. The summary should not just be a heap of related information, but should build from sentence to a coherent body of information about a topic."

Evaluation Steps:

1. Read the news article carefully and identify the main topic and key points.
2. Read the summary and compare it to the news article. Check if the summary covers the main topic and key points of the news article, and if it presents them in a clear and logical order.
3. Assign a score for coherence on a scale of 1 to 5, where 1 is the lowest and 5 is the highest based on the Evaluation Criteria.

Example:

Source Text:

{Document}

Summary:

{Summary}

Evaluation Form(scores ONLY): Make a selection from "1", "2", "3", "4", "5". Only write the answer with a single score, do not write reasons.

- Coherence:

## Prompts Used for Summeval for Consistency Evaluation

You will be given a news article. You will then be given one summary written for this article.

Your task is to rate the summary on one metric.

Please make sure you read and understand these instructions carefully. Please keep this document open while reviewing, and refer to it as needed.

Evaluation Criteria:

Consistency (1-5) - the factual alignment between the summary and the summarized source. A factually consistent summary contains only statements that are entailed by the source document. Annotators were also asked to penalize summaries that contained hallucinated facts.

Evaluation Steps:

1. Read the news article carefully and identify the main facts and details it presents.
2. Read the summary and compare it to the article. Check if the summary contains any factual errors that are not supported by the article.
3. Assign a score for consistency based on the Evaluation Criteria.

Example:

Source Text:

{Document}

Summary:

{Summary}

Evaluation Form(scores ONLY): Make a selection from "1", "2", "3", "4", "5". Only write the answer with a single score, do not write reasons.

- Consistency:

## Prompts Used for Summeval for Fluency Evaluation

You will be given one summary written for a news article.

Your task is to rate the summary on one metric.

Please make sure you read and understand these instructions carefully. Please keep this document open while reviewing, and refer to it as needed.

Evaluation Criteria:

Fluency (1-5): the quality of the summary in terms of grammar, spelling, punctuation, word choice, and sentence structure.

- 1: Poor. The summary has many errors that make it hard to understand or sound unnatural.
- 2: Below Average. The summary has several noticeable errors that significantly impact readability, though some parts can be understood with effort.
- 3: Fair. The summary has some errors that affect the clarity or smoothness of the text, but the main points are still comprehensible.
- 4: Good. The summary has minor errors that do not significantly interfere with understanding; it reads relatively smoothly.
- 5: Excellent. The summary has few or no errors and is easy to read and follow, with natural-sounding language throughout.

Example:

Summary:

{Summary}

Evaluation Form(scores ONLY): Make a selection from "1", "2", "3", "4", "5". Only write the answer with a single score, do not write reasons.

- Fluency:

## Prompts Used for Summeval for Relevance Evaluation

You will be given one summary written for a news article.

Your task is to rate the summary on one metric.

Please make sure you read and understand these instructions carefully. Please keep this document open while reviewing, and refer to it as needed.

Evaluation Criteria:

Relevance (1-5) - selection of important content from the source. The summary should include only important information from the source document. Annotators were instructed to penalize summaries which contained redundancies and excess information.

Evaluation Steps:

1. Read the summary and the source document carefully.
2. Compare the summary to the source document and identify the main points of the article.
3. Assess how well the summary covers the main points of the article, and how much irrelevant or redundant information it contains.
4. Assign a relevance score from 1 to 5.

Example:

Source Text:

{Document}

Summary:

{Summary}

Evaluation Form(scores ONLY): Make a selection from "1", "2", "3", "4", "5". Only write the answer with a single score, do not write reasons.

- Relevance:

**MT-Bench.** These prompts are reused from MT-Bench[43] with slight modifications.

---

### Prompts Used for MT-Bench

Human: For this task, you will be shown two conversations between a user and an AI assistant, labeled A and B. Your goal is to evaluate which response (A or B) better follows the user's instructions and more helpfully answers their question.

<PrefJudgment>
<Conversation A>
{conversation_a}
</Conversation A>

<Conversation B>
{conversation_b}
</Conversation B>

<Instructions>
Evaluate the two conversations and choose one of the following:
a: If Conversation A's AI assistant better follows the user's instructions and answers their question
b: If Conversation B's AI assistant better follows the user's instructions and answers their question
tie: If both AI assistants are equally good/poor in following instructions and answering the user's question

Consider factors like helpfulness, relevance, accuracy, depth, creativity, and appropriate level of detail when making your evaluation. Do not show positional bias towards A or B. Response length should not unduly influence your decision.
Make a selection from "a", "b", "tie". Only write the answer with a single word, do not write reasons.
</Instructions>
</PrefJudgment>

Assistant:

---

**NLI Tasks.** For SNLI and MNLI datasets, prompts are reused from LogiEval[51] with slight modifications.

---

**Prompts Used for NLI Tasks**

You will be given a premise and a hypothesis. Your task is to determine whether the hypothesis logically follows from the premise. Choose only one of the following labels and output your answer with ONLY the label (one word), ensuring there are no spaces or other characters in the answer.

Possible labels:

entailment: The hypothesis follows logically from the information contained in the premise.
neutral: It is not possible to determine whether the hypothesis is true or false without further information.
contradiction: The hypothesis is logically false from the information contained in the premise.

Read the following premise and hypothesis thoroughly and select the correct answer from the three answer labels.

Premise: {premise}

Hypothesis: {hypothesis}

Make a selection from "entailment", "neutral", "contradiction". Only write the answer with a single word, do not write reasons.

---

## G   Ethical Consideration

Our work explores the alignment of LLM-generated evaluation distributions with human judgment distributions, aiming to enhance the accuracy, diversity, and robustness of automatic evaluations. This has positive societal implications by potentially reducing the reliance on costly and time-consuming human evaluations, enabling scalable and fairer assessments in applications such as education, content moderation, and peer review. However, automated judgment systems also carry inherent risks. Misaligned or overconfident evaluations may lead to biased decisions, potentially reflecting and amplifying biases subtly present within the human data used for the alignment process itself. Such outcomes are particularly detrimental in high-stakes or subjective domains where fairness is paramount and the nuanced complexities of human judgment are not easily replicated or may be overlooked by automated systems.

