# OpenReview forum: "Distributional LLM-as-a-Judge"
_NeurIPS.cc/2025/Conference — NeurIPS 2025 poster_

### Official Review · Reviewer_3U8S · 2025-07-02

**Clarity:** 4
**Significance:** 3
**Originality:** 4
**Rating:** 5
**Confidence:** 3

**Summary:**

The paper addresses a key limitation in the LLM-as-a-Judge paradigm: human evaluations inherently involve diversity and uncertainty (reflected as distributions), but existing methods rely on single-point judgments, leading to information loss and reduced reliability. The authors propose a distribution alignment framework to align LLM-generated judgment distributions with empirical human evaluation distributions.  Extensive experiments demonstrate that that framework significantly outperforms existing closed-source LLMs and conventional single-point alignment methods, with improved alignment quality, evaluation accuracy, and robustness.

**Questions:**

Please refer to the comments above

**Ethical Concerns:**

["NO or VERY MINOR ethics concerns only"]

**Final Justification:**

I recommend accepting this paper, based on its novelty, technical strength, and evaluation comprehensiveness.

**Limitations:**

Yes

**Quality:**

4

**Strengths And Weaknesses:**

Strengths:

-The paper is novel in that it appears to be the first work to explicitly align LLM judgment distributions (not just point estimates) with human evaluation distributions.

- The proposed method is technically sound in that it introduces hybrid loss to balance distribution alignment and stability, as well as employing adversarial training to avoid overfitting to limited human annotations.

- The method is rigorously evaluated across diverse tasks and LLM backbones, followed by ablation studies to confirm the usefulness of each component.

Weaknesses:
- The approach requires datasets with multiple human annotations per sample to obtain empirical approximations, where are not commonly seen in practice.

- The approach may incur extra computational overhead in the introduced adversarial training strategy, but the overall efficiency appears undiscussed in the paper.

---

> ### Author Rebuttal · Authors · 2025-07-31
>
> Dear reviewer 3U8S,
>
> Thank you so much for your precious time and your positive feedback. We will now address your concerns.
>
> > **For Weakness 1: The approach requires datasets with multiple human annotations per sample to obtain empirical approximations, where are not commonly seen in practice.**
>
> Thank you for this valuable comment. We acknowledge that requiring multiple human annotations per sample is indeed a necessary condition for our method. However, we would like to offer the following points to alleviate this concern:
>
> 1. **Fundamental Importance:** As discussed in our paper, we believe that modeling the distribution of human judgments is fundamentally important. The natural disagreements among annotators carry valuable information that is often lost when reducing labels to a single point. This richness is particularly critical in **high-stakes domains**—such as medical diagnosis or policy-making—where capturing the full range of expert perspectives is essential. Therefore, we argue that the cost of collecting multiple human annotations is justified, given the significance of the problems we aim to address.
>
> 2. **Generalization Capability of Our Framework:** A key strength of modern LLMs is learning transferable knowledge. To verify that our framework learns a transferable representation of human disagreement, rather than overfitting to dataset-specific annotation patterns, we conducted an OOD experiment (training on SNLI, testing on the unseen ChaosNLI [1] dataset).
>
>    The results are presented in the table below, where we report mean ± std over several runs and `*` denotes a statistically significant improvement (p < 0.05). The data confirms our method's strong generalization to the unseen dataset. This suggests that the learned alignment capability is transferable, **potentially reducing the need to collect extensive multi-annotated data for every new task**.
>
>    | Model         | Method                  |     KL Divergence (↓)     |         Accuracy (↑)         |
>    | :------------ | :---------------------- | :-----------------------: | :--------------------------: |
>    | *Qwen2.5-7B*  | Raw model               |      $3.94_{±0.00}$       |      $60.3\\%_{±0.0\\%}$       |
>    |               | Single-point            |      $1.01_{±0.01}$       |      $66.5\\%_{±0.9\\%}$       |
>    |               | **Distribution (Ours)** | $\mathbf{0.31}^*_{±0.01}$ | $\mathbf{67.8\\%}^*_{±0.5\\%}$ |
>    | *LLaMA3.1-8B* | Raw model               |      $0.68_{±0.00}$       |      $57.8 \\%_{±0.0\\%}$      |
>    |               | Single-point            |      $1.15_{±0.05}$       |      $60.3\\%_{±0.1\\%}$       |
>    |               | **Distribution (Ours)** | $\mathbf{0.46}^*_{±0.02}$ | $\mathbf{62.7\\%}^*_{±0.1\\%}$ |
>
>
> 3. **Robustness with Limited Annotations:** In fact, during the early conceptualization of this work, we were fully aware of this potential limitation. To address this issue, our **adversarial training strategy** was specifically designed to enhance robustness, even when only a small number of annotations are available per sample. Our original experiments (Table 1 and 2) already demonstrate that our method achieves satisfactory performance under such settings.
>
>
>
> > **For Weakness 2: The approach may incur extra computational overhead in the introduced adversarial training strategy, but the overall efficiency appears undiscussed in the paper.**
>
> We thank the reviewer for this important practical point. The computational overhead is introduced by the inner PGD loop (K=5 steps), which finds the worst-case label distribution. Crucially, this PGD optimization computes gradients with respect to the label distribution `p(x)`, not the model parameters `θ`. During this inner loop, the model's output distribution `q_theta(x)` is treated as a fixed constant. As a result, these PGD steps are computationally inexpensive as they **do not involve any backpropagation through the main language model**.
>
> To empirically validate this, we conducted a direct comparison of the **training efficiency**, benchmarked on the MNLI dataset with the Qwen2.5-7B model on a single NVIDIA A100 GPU. The table below shows our method incurs a modest slowdown of approximately **21%**, which we believe is a worthwhile trade-off for the significant gains in alignment quality.
>
>
> | Method                  | Time / Epoch (min) | Throughput (samples/sec) | Relative Slowdown |
> | :---------------------- | :----------------: | :----------------------: | :---------------: |
> | Single-point            |       $23.4$       |          $5.70$          |   $1.0 \times$    |
> | **Distribution (Ours)** |       $28.3$       |          $4.71$          |   $1.21\times$    |
>
> We will add a discussion of this analysis and include these results in the final version of our paper.
>
>
> **We sincerely thank you for your time and insightful comments, which are very important for improving our paper. We hope our responses have addressed your concerns.**
>
> **References:**
>
> [1] Nie, Y., et al. "What can we learn from collective human opinions on natural language inference data." *EMNLP* (2020): 9131-9143.

---

> > ### Author Response · Authors · 2025-08-05
> >
> > Dear Reviewer 3U8S,
> >
> >
> > Thank you so much for your strong support for our work. We are greatly encouraged by your positive and insightful review.  We hope our response has fully addressed your concerns, and we are happy to discuss any further points you may have. Thank you again for your time and for helping to improve our paper.
> >
> >
> >
> >  Best regards,
> >
> >  Authors of Submission 26507

---

> > ### Comment · Reviewer_3U8S · 2025-08-09
> >
> > Thanks for your response. I will keep my score as is.

---

### Official Review · Reviewer_6xen · 2025-07-02

**Clarity:** 3
**Significance:** 3
**Originality:** 2
**Rating:** 5
**Confidence:** 3

**Summary:**

The current LLM-as-a-Judge setup relies on single-point judgments which dismisses the inherent subjectivity of certain tasks and inputs. Instead, this paper aims to have a judgment distribution as an output of this paradigm.

The paper proposes a framework that aligns the output distribution of an LLM with the human judgment distributions. This is done through a hybrid loss-function that uses both KL divergence to minimize the distance. For improved robustness, reducing the overfitting to dataset specific noise, the authors apply a worst-case perturbation to the distributions. This is done through Projected Gradient Descent.

The framework is tested on Natural Language Inference (MNLI and SNLI), Qualitative Assessment, and Human Preference Understanding, using both open-source (Qwen2.5-7B and LLaMa3.1-8B) and closed-source (GPT4o, GPT4o-mini) models. To evaluate the framework, accuracy and KL-divergence is used. The results indicate that the proposed framework outperforms other baselines in terms of KL-divergence and yields competitive results for accuracy.

**Questions:**

- Why do you opt for the KL loss specifically and not for example Jensen-Shannon Divergence, which is a symmetrical version of it?
- Line 204-205: I think you accidentally mixed up closed-source and open-source in the sentence.
- Shouldn’t perturbation take into account annotator information? For instance, if there is no high diversity in annotators, the perturbation could be higher? Or is that impractical for the real world? (I am just curious about your thoughts on this specific question)
- Line 242 - 245: Doesn’t gpt-4o-mini do better than gpt-4o in terms of accuracy for the NLI datasets

**Ethical Concerns:**

["NO or VERY MINOR ethics concerns only"]

**Final Justification:**

I think the paper addresses an important problem of the potential subjectivity of certain tasks with LLM-as-a-judge.

The proposed framework is rigorously tested and evaluated accordingly. I am still a bit apprehensive about using the hard CE loss for a task on subjectivity, intuitively this does not seem very straightforward to me, however, the authors backup this choice with empirical results indicating that this provides stability during training.

They cleared up my doubts about using accuracy as a metric, which is a secondary metric for verification but the focus is more on KL-divergence.

The authors added my suggested experiments, including statistical tests, measuring performance on out-of-distribution datasets, experimenting with Jensen Shannon Divergence as a loss function and testing the framework on ChaosNLI. This strengthens the findings of the authors and I think the subfield of subjective modeling can benefit from the proposed method which goes beyond mere prompting strategies.

**Limitations:**

yes

**Quality:**

3

**Strengths And Weaknesses:**

**Strengths**
- The paper addresses an important limitation of the current LLM-as-a-Judge paradigm. It is important that such a judge can reflect multiple viewpoints, especially for tasks that include a lot of subjectivity.
- The paper is well-written and structured and has an easy to follow flow.
- Good analyses through ablation studies, e.g., looking at the impact of both the KL and CE losses and the hyperparameters.

**Weaknesses**
- While the authors are working toward a model that can output the distribution, I find some specific design choices in the setup rather contradicting of the notion. Firstly, I wonder if it is a good idea to incorporate the aggregated majority vote in the loss function through the CE loss (I am assuming this is the hard version of the loss function and not the soft CE loss, please correct me if I am wrong). If we have a rather spread out distribution, then there is no clear single ground-truth label. Consequently, this reflects into the usage of accuracy as a metric. Though I admit that measuring correctness gets a bit though to measure in such a case but I think at least a discussion around this would strengthen the paper a lot.
- I wonder why ChaosNLI [1] was not used for the experiments. This is an NLI dataset with 100 annotations per sample, which would be a great fit for the experiments conducted in this paper.
- Some analysis of the method’s robustness to other unseen datasets would have been nice. E.g., the models for SNLI and MNLI could have been tested on ChaosNLI.
- Given that some of the models show very similar accuracy scores, statistical significance tests would be really useful.

---------------------------------------
[1] Yixin Nie, Xiang Zhou, and Mohit Bansal. 2020. What Can We Learn from Collective Human Opinions on Natural Language Inference Data?. In Proceedings of the 2020 Conference on Empirical Methods in Natural Language Processing (EMNLP), pages 9131–9143, Online. Association for Computational Linguistics.

---

> ### Author Rebuttal · Authors · 2025-07-31
>
> Dear reviewer 6xen,
>
> Thank you so much for your precious time and your thoughtful feedback. We will now address your concerns.
>
> > **For Weakness 1: The using of CE loss and accuracy metric may contradict our paper's goal of modeling distributions.**
>
> We thank the reviewer for this fundamental question. These designs are not contradictory, but are deliberate and principled decisions to ensure both **training stability** and **meaningful evaluation**.
>
> **1. CE loss -- for learning stability:**
>
> The reviewer is correct that we use a standard CE loss on the majority-vote label. However, its role is not as a primary objective, but as an **auxiliary regularizer for training stability**. As we noted in our paper (Line 129-133), this design is directly inspired by the Knowledge Distillation [2, 3] framework, where a "hard" label loss is used to stabilize learning from a "soft" target distribution. Our own results in Figure 3 empirically validate this necessity: on challenging datasets like MT-Bench, a pure KL objective (`α=1.0`) is unstable and performs worse, demonstrating the crucial role of the CE term as a stabilizing anchor.
>
> **2. Accuracy metric -- for meaningful evaluation:**
>
> Similarly, **Accuracy is a secondary, complementary metric**, with KL divergence being our primary measure of success. We include it for two practical reasons: it serves as a valuable indicator of our model's ability to capture the majority consensus, and more critically, it provides a crucial bridge for comparison with prior work that relies solely on this measure. This allows us to demonstrate that we achieve our primary goal without sacrificing performance on this traditional standard.
>
> We will add a more detailed discussion of this rationale to the paper. We thank the reviewer again for this valuable prompt.
>
>
>
> > **For Weakness 2: Not using the highly relevant ChaosNLI [1] dataset.**
>
> We sincerely thank the reviewer for this important suggestion. We agree that ChaosNLI is an ideal and challenging benchmark for our distribution alignment framework. Consequently, we have conducted **a new set of experiments on the ChaosNLI dataset**. The results are presented in the table below, where we report **mean ± std** over multiple runs. The `*` indicates that the improvement of our method over the single-point baseline is statistically significant (p < 0.05).
>
> | Model | Method | KL Divergence (↓) | Accuracy (↑) |
> | :--- | :--- | :---: | :---: |
> | *GPT-4o-mini* | Raw model | $3.92_{±0.00}$ | $64.1\\%_{±0.0\\%}$ |
> | *GPT-4o* | Raw model | $2.43_{±0.00}$ | $61.2\\%_{±0.0\\%}$ |
> | *Qwen2.5-7B* | Raw model | $3.94_{±0.00}$ | $60.3\\%_{±0.0\\%}$ |
> | | Single-point | $1.22_{±0.02}$ | $70.6\\%_{±0.7\\%}$ |
> | | **Distribution (Ours)** | $\mathbf{0.41}^*_{±0.02}$ | $\mathbf{71.8\\%}^*_{±0.5\\%}$ |
> | *LLaMA3.1-8B* | Raw model | $0.68_{±0.00}$ | $57.8\\%_{±0.0\\%}$ |
> | | Single-point | $1.14_{±0.04}$ | $65.0\\%_{±0.5\\%}$ |
> | | **Distribution (Ours)** | $\mathbf{0.43}^*_{±0.05}$ | $65.7\\%_{±1.2\\%}$ |
>
> The results on ChaosNLI are fully consistent with our original findings: our method significantly outperforms baselines in KL divergence while maintaining competitive accuracy. We will incorporate these results into the final paper and are grateful for the valuable suggestion.
>
>
>
> > **For Weakness 3: Analyze the method’s robustness to other unseen datasets.**
>
> We thank the reviewer for this excellent suggestion. While our paper already demonstrated robustness to label noise (Sec 4.5), we agree that testing for **out-of-distribution (OOD) generalization** is a complementary and even more challenging evaluation. Therefore, we conducted a new OOD experiment, fine-tuning models on the SNLI training set and evaluating them directly on the **unseen ChaosNLI** test set:
>
> | Model         | Method                  |     KL Divergence (↓)     |         Accuracy (↑)         |
> | :------------ | :---------------------- | :-----------------------: | :--------------------------: |
> | *Qwen2.5-7B*  | Raw model               |      $3.94_{±0.00}$       |      $60.3\\%_{±0.0\\%}$       |
> |               | Single-point            |      $1.01_{±0.01}$       |      $66.5\\%_{±0.9\\%}$       |
> |               | **Distribution (Ours)** | $\mathbf{0.31}^*_{±0.01}$ | $\mathbf{67.8\\%}^*_{±0.5\\%}$ |
> | *LLaMA3.1-8B* | Raw model               |      $0.68_{±0.00}$       |      $57.8 \\%_{±0.0\\%}$      |
> |               | Single-point            |      $1.15_{±0.05}$       |      $60.3\\%_{±0.1\\%}$       |
> |               | **Distribution (Ours)** | $\mathbf{0.46}^*_{±0.02}$ | $\mathbf{62.7\\%}^*_{±0.1\\%}$ |
>
> These OOD results demonstrate our method's superior generalization over the baseline, confirming that our framework learns a transferable representation of human disagreement instead of merely overfitting to dataset-specific patterns. We will integrate this crucial analysis into our final paper and are grateful for the valuable suggestion.
>
>
>
> > **For Weakness 4: The need for statistical significance tests.**
>
> We thank the reviewer for this crucial point on experimental rigor. We completely agree that statistical testing is essential, particularly for metrics with small margins like accuracy. We have conducted new **multi-run experiments (N=5)**, focusing specifically on the datasets with the narrowest accuracy margins: **SNLI and MNLI**.
>
> | Model         | Method                  |        SNLI KL (↓)        |         SNLI Acc (↑)         |        MNLI KL (↓)        |         MNLI Acc (↑)         |
> | :------------ | :---------------------- | :-----------------------: | :--------------------------: | :-----------------------: | :--------------------------: |
> | *Qwen2.5-7B*  | Single-point            |      $0.60_{±0.01}$       |      $92.7\\%_{±0.1\\%}$       |      $0.64_{±0.02}$       |      $89.7\\%_{±0.2\\%}$       |
> |               | **Distribution (Ours)** | $\mathbf{0.23}^*_{±0.01}$ | $\mathbf{93.3\\%}^*_{±0.2\\%}$ | $\mathbf{0.23}^*_{±0.00}$ |      $89.8\\%_{±0.2\\%}$       |
> | *LLaMA3.1-8B* | Single-point            |      $0.69_{±0.02}$       |      $92.4\\%_{±0.42\\%}$      |      $0.67_{±0.02}$       |      $89.6\\%_{±0.2\\%}$       |
> |               | **Distribution (Ours)** | $\mathbf{0.28}^*_{±0.01}$ |      $92.4\\%_{±0.13\\%}$      | $\mathbf{0.24}^*_{±0.02}$ | $\mathbf{90.0\\%}^*_{±0.2\\%}$ |
>
> The table highlights our method's key achievement: yielding substantial and statistically significant improvements in KL divergence, while maintaining competitive accuracy. While these multi-run experiments were limited to SNLI/MNLI due to time constraints, we will include these rigorous statistics for all datasets in the camera-ready version. Thank you again for this feedback.
>
>
>
> > **For Q1: Why not using JS loss?**
>
> We thank the reviewer for this insightful question. Our choice of KL divergence is based on two main reasons:
>
> - First, our choice is motivated by its established success as the standard measure in paradigms like Knowledge Distillation [2, 3], which shares the core technical goal of aligning a model's output distribution with a target "soft" distribution.
> - Second, KL divergence generally offers more stable gradients for optimization compared to JS divergence, which can suffer from vanishing gradients. [4]
>
> To empirically validate this choice, we conducted a new experiment comparing both training objectives on SNLI using Qwen2.5-7B. The results below confirm that using KL divergence yields stronger performance across all metrics:
>
> | Method | KL Divergence (↓) | JS Divergence (↓) | Accuracy (↑) |
> | :--- | :---: | :---: | :---: |
> | **Ours (using KL Div.)** | $\mathbf{0.23}^*_{±0.00}$ | $\mathbf{0.049}^*_{±0.000}$ | $\mathbf{93.2\\%}^*_{±0.2\\%}$ |
> | Ours (using JS Div.) | $0.27_{±0.00}$ | $0.050_{±0.000}$ | $92.1\\%_{±0.2\\%}$ |
>
>
>
> > **For Q2: Typo on Line 204-205.**
>
> We thank the reviewer for catching this error. We will correct this typo in the final version.
>
>
>
> > **For Q3: Perturbation taking into account annotator information.**
>
> This is a fascinating and thought-provoking question. We agree with the reviewer's excellent intuition: if the annotator pool lacks diversity, the collected distribution may be biased, and a larger perturbation radius `ε` would be a principled way to account for this unobserved variance. The primary practical obstacle, however, is the lack of annotator metadata in most crowdsourced datasets, and we would like to leave this direction as an important future work.
>
>
>
> >**For Q4: Accuracy of GPT-4o-mini vs. GPT-4o on NLI tasks**
>
> Thank you for this sharp observation. GPT-4o-mini indeed achieves slightly higher accuracy on the NLI datasets than GPT-4o. Our hypothesis is that on a mature and relatively well-defined task like NLI, the accuracy of powerful models can be very close.
>
>
>
> **We sincerely thank you for your time and insightful comments, which are very important for improving our paper. We hope our responses have addressed your concerns.**
>
>
>
> **References:**
>
> [1] Nie, Y., et al. "What can we learn from collective human opinions on natural language inference data." *EMNLP* (2020): 9131-9143.
>
> [2] Hinton, Geoffrey, et al. "Distilling the knowledge in a neural network." *arXiv:1503.02531* (2015).
>
> [3] Sanh, Victor, et al. "DistilBERT, a distilled version of BERT: smaller, faster, cheaper and lighter." *arXiv:1910.01108* (2019).
>
> [4] Arjovsky, Martin, et al. "Wasserstein GAN." *PMLR* (2017): 214-223.

---

> > ### Author Response · Authors · 2025-08-05
> >
> > Dear Reviewer 6xen,
> >
> > As a follow-up to our rebuttal, and to provide the most complete information for the final discussion, we are pleased to share the completed multi-run experiment results for our remaining datasets, SummEval and MT-Bench.
> >
> > The updated main results table, now with `mean ± std` and significance tests, is presented below.
> >
> > | Model         | Method                  |      Summeval KL (↓)      |       Summeval Acc (↑)       |      MT-Bench KL (↓)      | MT-Bench Acc (↑)  |
> > | :------------ | :---------------------- | :-----------------------: | :--------------------------: | :-----------------------: | :---------------: |
> > | *Qwen2.5-7B*  | Single-point            |      $0.73_{±0.03}$       |      $45.6\\%_{±0.5\\%}$       |      $0.82_{±0.01}$       | $64.0\\%_{±1.0\\%}$ |
> > |               | **Distribution (Ours)** | $\mathbf{0.53}^*_{±0.02}$ |      $45.9\\%_{±0.4\\%}$       | $\mathbf{0.68}^*_{±0.02}$ | $65.4\\%_{±1.4\\%}$ |
> > | *LLaMA3.1-8B* | Single-point            |      $0.67_{±0.05}$       |      $45.7\\%_{±0.8\\%}$       |      $0.81_{±0.01}$       | $62.1\\%_{±0.8\\%}$ |
> > |               | **Distribution (Ours)** | $\mathbf{0.51}^*_{±0.01}$ | $\mathbf{47.3^*\\%}_{±0.6\\%}$ | $\mathbf{0.74}^*_{±0.01}$ | $62.8\\%_{±0.8\\%}$ |
> >
> > This complete table confirms our method's key achievement: delivering statistically significant improvements in distributional alignment (KL divergence), while maintaining competitive accuracy across all original datasets.
> >
> > **We hope this additional data is helpful for the final evaluation and are happy to answer any further questions.**
> >
> > Best regards,
> >
> > Authors of Submission 26507

---

> > ### Comment · Reviewer_6xen · 2025-08-06
> >
> > Thank you for addressing my concerns and running the suggested experiments! These really strengthen the findings of the paper.
> >
> > I have thus decided to raise my score.

---

> > > ### Author Response · Authors · 2025-08-06
> > >
> > > Thank you so much for your encouraging and positive feedback. We are truly grateful for your dedicated time and for providing the constructive comments that have helped strengthen our paper!

---

### Official Review · Reviewer_a38u · 2025-07-03

**Clarity:** 4
**Significance:** 3
**Originality:** 4
**Rating:** 4
**Confidence:** 4

**Summary:**

This work introduces a framework for aligning an LLM-based judge's output distribution with that of human annotators. The frameworks include a hybrid KL + CE loss and adversarial training on perturbed inputs via PGD. Evaluation on 4 tasks and 2 models shows the effectiveness of the proposed framework.

**Questions:**

1. [3] should be cited in Section 4. The formulations are identical: a CE + KL objective solved in a PGD min–max loop, only this work shifts the KL term from input perturbations to label perturbations.
2. Can this approach generalize to unseen datasets, e.g., with a different label space?
3. In Figure 3, why does the KL divergence increase considerably at alpha=1.0 for MTBench, but none of the other tasks? Is there some feature of MTBench that necessitates the CE term more so than the other datasets?

[3] Zhang, Hongyang, et al. "Theoretically principled trade-off between robustness and accuracy." International conference on machine learning. PMLR, 2019.

I'm willing to increase my score if the evaluation is improved.

**Ethical Concerns:**

["NO or VERY MINOR ethics concerns only"]

**Final Justification:**

The authors presented a series of new experiments that resolved many of my concerns and significantly improved the quality of the evaluation. While the reduction in KL divergence is impressive, I'm not convinced the required annotation cost is justified. Therefore, I'm giving a borderline score of 4 and leaving the decision to the AC.

**Limitations:**

Yes

**Quality:**

3

**Strengths And Weaknesses:**

Strengths:
1. The paper is well-organized and reads well.
2. The approach is simple yet principled. Figures 1 and 2 are clear and aid in understanding.
3. Evaluation reveals the method is effective, with consistent improvements in KL divergence.

Weaknesses:
1. Despite the reduction in KL divergence, the improvements in accuracy are minimal. As such, more motivation is needed to justify modeling the full distribution given the added data annotation cost.
2. The evaluation is underwhelming. First, the margins (particularly accuracy) are slim, and only point estimates are provided. Second, the dataset choice could be improved to better fit the LLM-as-a-Judge narrative (e.g., [1] and [2], both have ~3 annotations per instance). Lastly, the method is only tested on in-distribution tasks so the generaliability of the framework is not clear.

[1] Wang, Zhilin, et al. "Helpsteer2: Open-source dataset for training top-performing reward models." arXiv preprint arXiv:2406.08673 (2024).

[2] Wang, Zhilin, et al. "HelpSteer3-Preference: Open Human-Annotated Preference Data across Diverse Tasks and Languages." arXiv preprint arXiv:2505.11475 (2025).

---

> ### Author Rebuttal · Authors · 2025-07-31
>
> Dear Reviewer a38u,
>
> Thank you for your thoughtful feedback. We will address your concerns below.
>
> > **For Weakness 1: Despite the reduction in KL divergence, the improvements in accuracy are minimal. As such, more motivation is needed to justify modeling the full distribution given the added data annotation cost.**
>
> The core value of our work extends beyond improving single-point accuracy. Instead, our goal is to capture the rich, underlying structure of human evaluations—such as the variance of opinions, which signals the level of consensus. This distributional information is valuable in high-stakes applications like medical diagnosis, where understanding the full spectrum of expert opinions is more useful and safer signal than a single, overconfident prediction.
>
> Accordingly, our evaluation is centered on KL Divergence as the primary metric, because it directly quantifies how well our model captures this entire information-rich distribution. Accuracy serves as a secondary metric to ensure our method captures the majority consensus and remains competitive with traditional standards.
>
> We will expand on this motivation in the final paper.
>
>
>
> > **For Weakness 2: The evaluation is underwhelming. First, the margins (particularly accuracy) are slim, and only point estimates are provided. Second, the dataset choice could be improved to better fit the LLM-as-a-Judge narrative. Lastly, the method is only tested on in-distribution tasks so the generaliability of the framework is not clear.**
>
> **1. On slim margins and point estimates**
>
> We have conducted new multi-run experiments (N=5) on the datasets with the narrowest margins: **SNLI and MNLI**. The results are reported in the table below, where we report **mean ± std** over multiple runs. The `*` indicates that the improvement of our method over the single-point baseline is statistically significant (p < 0.05).
>
> | Model         | Method                  |        SNLI KL (↓)        |         SNLI Acc (↑)         |        MNLI KL (↓)        |         MNLI Acc (↑)         |
> | :------------ | :---------------------- | :-----------------------: | :--------------------------: | :-----------------------: | :--------------------------: |
> | *Qwen2.5-7B*  | Single-point            |      $0.60_{±0.01}$       |      $92.7\\%_{±0.1\\%}$       |      $0.64_{±0.02}$       |      $89.7\\%_{±0.2\\%}$       |
> |               | **Distribution (Ours)** | $\mathbf{0.23}^*_{±0.01}$ | $\mathbf{93.3\\%}^*_{±0.2\\%}$ | $\mathbf{0.23}^*_{±0.00}$ |      $89.8\\%_{±0.2\\%}$       |
> | *LLaMA3.1-8B* | Single-point            |      $0.69_{±0.02}$       |      $92.4\\%_{±0.42\\%}$      |      $0.67_{±0.02}$       |      $89.6\\%_{±0.2\\%}$       |
> |               | **Distribution (Ours)** | $\mathbf{0.28}^*_{±0.01}$ |      $92.4\\%_{±0.13\\%}$      | $\mathbf{0.24}^*_{±0.02}$ | $\mathbf{90.0\\%}^*_{±0.2\\%}$ |
>
> The results highlight our key achievement: yielding substantial and statistically significant improvements in KL divergence, while maintaining competitive accuracy. We will include these rigorous statistics for all datasets in the final version.
>
> **2. On the choice of datasets**
>
> We have conducted new experiments on these modern benchmarks (**HelpSteer2** [1] for Qualitative Assessment task, and **HelpSteer2-Preference** [2] for Human Preference Understanding task). The results for both datasets are presented below.
>
> **HelpSteer2**
>
> | Model | Method | KL Divergence (↓) | Accuracy (↑) |
> | :--- | :--- | :---: | :---: |
> | *GPT-4o-mini* | Raw model | $4.83_{±0.00}$ | $42.4\\%_{±0.0\\%}$ |
> | *GPT-4o* | Raw model | $2.09_{±0.00}$ | $40.4\\%_{±0.0\\%}$ |
> | *Qwen2.5-7B* | Raw model | $3.79_{±0.00}$ | $32.0\\%_{±0.0\\%}$ |
> | | Single-point | $0.76_{±0.03}$ | $60.5\\%_{±0.1\\%}$ |
> | | **Distribution (Ours)** | $\mathbf{0.63}^*_{±0.01}$ | $60.0\\%_{±0.5\\%}$ |
> | *LLaMA3.1-8B* | Raw model | $2.50_{±0.00}$ | $13.3\\%_{±0.0\\%}$ |
> | | Single-point | $0.73_{±0.02}$ | $62.4\\%_{±0.3\\%}$ |
> | | **Distribution (Ours)** | $\mathbf{0.59}^*_{±0.00}$ | $62.4\\%_{±0.1\\%}$ |
>
> **HelpSteer2-Preference**
>
> | Model | Method | KL Divergence (↓) | Accuracy (↑) |
> | :--- | :--- | :---: | :---: |
> | GPT-4o-mini | Raw model | $13.8_{±0.00}$ | $9.2\\%_{±0.0\\%}$ |
> | GPT-4o | Raw model | $4.98_{±0.00}$ | $18.9\\%_{±0.0\\%}$ |
> | *Qwen2.5-7B* | Raw model | $7.65_{±0.00}$ | $10.0\\%_{±0.0\\%}$ |
> | | Single-point | $0.57_{±0.02}$ | $71.4\\%_{±0.8\\%}$ |
> | | **Distribution (Ours)** | $\mathbf{0.49}^*_{±0.01}$ | $71.3\\%_{±1.1\\%}$ |
> | *LLaMA3.1-8B* | Raw model | $2.86_{±0.00}$ | $14.9\\%_{±0.0\\%}$ |
> | | Single-point | $0.51_{±0.01}$ | $71.6\\%_{±0.5\\%}$ |
> | | **Distribution (Ours)** | $\mathbf{0.47}^*_{±0.00}$ | $\mathbf{73.8\\%}^*_{±0.3\\%}$ |
>
> The results on both modern benchmarks provide strong additional evidence for our framework's effectiveness. We will incorporate these extensive new results into the final paper and are grateful for this suggestion.
>
> **3. On out-of-distribution tasks and generalizability**
>
> We thank the reviewer for this excellent suggestion. While our paper already demonstrated robustness to label noise (Sec 4.5), we agree that testing for out-of-distribution (OOD) generalization is a complementary and more challenging evaluation. Therefore, we conducted a new OOD experiment, fine-tuning models on SNLI and evaluating them directly on the **unseen** ChaosNLI [3] dataset. The results are presented below.
>
> | Model         | Method                  |     KL Divergence (↓)     |         Accuracy (↑)         |
> | :------------ | :---------------------- | :-----------------------: | :--------------------------: |
> | *Qwen2.5-7B*  | Raw model               |      $3.94_{±0.00}$       |      $60.3\\%_{±0.0\\%}$       |
> |               | Single-point            |      $1.01_{±0.01}$       |      $66.5\\%_{±0.9\\%}$       |
> |               | **Distribution (Ours)** | $\mathbf{0.31}^*_{±0.01}$ | $\mathbf{67.8\\%}^*_{±0.5\\%}$ |
> | *LLaMA3.1-8B* | Raw model               |      $0.68_{±0.00}$       |      $57.8 \\%_{±0.0\\%}$      |
> |               | Single-point            |      $1.15_{±0.05}$       |      $60.3\\%_{±0.1\\%}$       |
> |               | **Distribution (Ours)** | $\mathbf{0.46}^*_{±0.02}$ | $\mathbf{62.7\\%}^*_{±0.1\\%}$ |
>
> These OOD results demonstrate our method's superior generalization over the baseline, confirming that our framework learns a transferable representation of human disagreement. We will integrate this crucial analysis into our final paper and are grateful for the valuable suggestion.
>
>
>
> > **For Q1: [3] should be cited in Section 4. The formulations are identical: a CE + KL objective solved in a PGD min–max loop, only this work shifts the KL term from input perturbations to label perturbations.**
>
> We thank the reviewer for their sharp observation and the opportunity to clarify this connection.
>
> While there is a structural resemblance in the optimization form, the two works operate in fundamentally different domains with distinct goals. TRADES addresses robustness to external input attacks in computer vision. Our work, in contrast, addresses distributional alignment with human opinions in the paradigm of LLM-as-a-Judge.
>
> This fundamental difference in goals is directly reflected in our loss design. In TRADES, the KL term is an **auxiliary regularizer** to enforce output smoothness against input perturbations. In our framework, the KL term is the **primary objective**, designed to directly align model outputs with the target human distribution. Our auxiliary CE loss merely serves to stabilize this primary alignment task, a design validated by our hyper-parameter analysis.
>
> We will revise our manuscript to cite TRADES and add this important contextual discussion. Thank you again for this valuable prompt.
>
> > **For Q2: Can this approach generalize to unseen datasets, e.g., with a different label space?**
>
>  As demonstrated in our new OOD experiment (presented in our response to Weakness 2.3), our framework generalizes well. The results confirm that our distribution alignment method's capabilities transfer to the unseen ChaosNLI dataset significantly better than the single-point baseline.
>
> > **For Q3: In Figure 3, why does the KL divergence increase considerably at alpha=1.0 for MTBench, but none of the other tasks? Is there some feature of MTBench that necessitates the CE term more so than the other datasets?**
>
> This is a sharp observation. Our hypothesis is that this phenomenon is tied to the **high difficulty and subjectivity** of certain tasks. The greater disagreement among annotators leads to more complex target distributions `p(x)`. For such challenging distributions, a pure KL divergence objective (`α=1.0`) can be unstable.
>
> To verify this hypothesis, we conducted the same analysis on **HelpSteer2-Preference**. Similar to MT-Bench, this is a benchmark for subjective human preference. We observed the exact same pattern—a **sharp increase** in KL divergence when the CE loss was completely removed:
>
> | Alpha ($\alpha$)      | 0.0  | 0.2  | 0.4  | 0.6  | 0.8      | 1.0      |
> | --------------------- | ---- | ---- | ---- | ---- | -------- | -------- |
> | **KL Divergence (↓)** | 0.57 | 0.53 | 0.50 | 0.49 | **0.48** | **2.01** |
>
> The consistent results from both datasets show that the auxiliary CE loss is crucial for stabilizing these difficult alignment tasks.
>
>
>
> **We sincerely thank you for your time and insightful comments, which are very important for improving our paper. We hope our responses have addressed your concerns.**
>
>
>
> **References:**
>
> [1] Wang, Zhilin, et al. "Helpsteer2: Open-source dataset for training top-performing reward models." *arXiv:2406.08673* (2024).
>
> [2] Wang, Zhilin, et al. "HelpSteer2-Preference: Complementing Ratings with Preferences." *arXiv:2410.01257* (2024).
>
> [3] Nie, Y., et al. "What can we learn from collective human opinions on natural language inference data." *EMNLP* (2020): 9131-9143.

---

> > ### Comment · Reviewer_a38u · 2025-08-05
> >
> > Thank you for the response. I have updated my score.

---

> > > ### Author Response · Authors · 2025-08-05
> > >
> > > Thank you for re-evaluating our work. We are truly grateful for your dedicated time and constructive comments throughout the review process.

---

### Official Review · Reviewer_Lp1a · 2025-07-23

**Clarity:** 3
**Significance:** 2
**Originality:** 2
**Rating:** 4
**Confidence:** 4

**Summary:**

The authors investigate the effectiveness of training LLM-as-a-Judge models using distributions of human judgments rather than majority vote aggregations. They use a hybrid loss objective which combines KL divergence between the two distribution of labels with a deterministic label. This method shows better performance on four standard benchmarks.

**Questions:**

- Highly relevant paper not yet cited: https://arxiv.org/abs/2104.0867, which discusses exactly this problem in the context of NLI, which you also use. Given the identical task definition, the non-mention of this or other related works is a substantial weakness for the paper.
- If you set a dataset size budget (number of human judgments collected), would it be more effective to sample several judgements for a single datapoint (to provide a distribution to train on) or new examples? I worry that this method is only effective due to essentially training on a much larger dataset than single-point. If correcting for this, which method performs better?

**Ethical Concerns:**

["NO or VERY MINOR ethics concerns only"]

**Final Justification:**

The rebuttal provided much appreciated experimental results and rigor to the paper, and improved my understanding of the significance of the results.

**Limitations:**

- Not many alternative methods are compared, related works section is very limited and missing key citations (see above)
- Limited experimental investigation
- Method requires high-redundancy annotations, which are costly.

**Paper Formatting Concerns:**

- The paper is well formatted and the figures are clear

**Quality:**

3

**Strengths And Weaknesses:**

Strengths
- Achieves best performance on several tasks as compared to previous methods
- Introduces a new adversarial loss objective for the alignment with human distributions that improves performance
- Discussion of hyperparameter impact is appreciated

Weaknesses
- No evaluation on tasks more similar to what LLM-as-a-Judge is often used for in practice, such as dataset labeling, quality evaluation, or pairwise preference prediction. That stated, the chosen subset of tasks (NLI, summarization, dialogue evaluation) is reasonable enough for research. The paper could be improved by expanding the tasks for which LLM-as-a-Judge is used to more closely align with common use-cases or incoporate a more diverse range of tasks.
- Limited experimental results and minimal evaluation. There is essentially only a single result in the paper: better performance than training on single-point data or raw LLM judges.

---

> ### Author Rebuttal · Authors · 2025-07-31
>
> Dear Reviewer Lp1a,
> Thank you for your rigorous and constructive review. We present our detailed responses below.
>
> > **For Weakness 1: No evaluation on tasks more similar to what LLM-as-a-Judge is often used for in practice, ... incoporate a more diverse range of tasks.**
>
> Thank you for the feedback on practical relevance. We find that some of our chosen tasks correspond well to the paradigms the reviewer mentioned: SummEval is a 'quality evaluation' task, and MT-Bench addresses 'pairwise preference prediction'.
>
> To further strengthen our evaluation, we have conducted new experiments on the recently proposed datasets, which align more closely with common use-cases. **HelpSteer2** [1] provides a state-of-the-art benchmark for 'quality evaluation' of modern LLM generations, while **HelpSteer2-Preference** [2] offers a more fine-grained benchmark for 'preference prediction' on paired LLM responses. The results are presented below, where we report **mean ± std** over multiple runs. The `*` indicates that the improvement of our method over the single-point baseline is statistically significant (p < 0.05).
>
> **HelpSteer2**
>
> | Model         | Method                  |     KL (↓)     |   Accuracy (↑)    |
> | :------------ | :---------------------- | :-----------------------: | :---------------: |
> | *GPT-4o-mini* | Raw model               |      $4.83_{±0.00}$       | $42.4\\%_{±0.0\\%}$ |
> | *GPT-4o*      | Raw model               |      $2.09_{±0.00}$       | $40.4\\%_{±0.0\\%}$ |
> | *Qwen2.5-7B*  | Raw model               |      $3.79_{±0.00}$       | $32.0\\%_{±0.0\\%}$ |
> |               | Single-point            |      $0.76_{±0.03}$       | $60.5\\%_{±0.1\\%}$ |
> |               | **Distribution (Ours)** | $\mathbf{0.63}^*_{±0.01}$ | $60.0\\%_{±0.5\\%}$ |
> | *LLaMA3.1-8B* | Raw model               |      $2.50_{±0.00}$       | $13.3\\%_{±0.0\\%}$ |
> |               | Single-point            |      $0.73_{±0.02}$       | $62.4\\%_{±0.3\\%}$ |
> |               | **Distribution (Ours)** | $\mathbf{0.59}^*_{±0.00}$ | $62.4\\%_{±0.1\\%}$ |
>
> **HelpSteer2-Preference**
>
> | Model         | Method                  |     KL (↓)     |         Accuracy (↑)         |
> | :------------ | :---------------------- | :-----------------------: | :--------------------------: |
> | GPT-4o-mini   | Raw model               |      $13.8_{±0.00}$       |       $9.2\\%_{±0.0\\%}$       |
> | GPT-4o        | Raw model               |      $4.98_{±0.00}$       |      $18.9\\%_{±0.0\\%}$       |
> | *Qwen2.5-7B*  | Raw model               |      $7.65_{±0.00}$       |      $10.0\\%_{±0.0\\%}$       |
> |               | Single-point            |      $0.57_{±0.02}$       |      $71.4\\%_{±0.8\\%}$       |
> |               | **Distribution (Ours)** | $\mathbf{0.49}^*_{±0.01}$ |      $71.3\\%_{±1.1\\%}$       |
> | *LLaMA3.1-8B* | Raw model               |      $2.86_{±0.00}$       |      $14.9\\%_{±0.0\\%}$       |
> |               | Single-point            |      $0.51_{±0.01}$       |      $71.6\\%_{±0.5\\%}$       |
> |               | **Distribution (Ours)** | $\mathbf{0.47}^*_{±0.00}$ | $\mathbf{73.8\\%}^*_{±0.3\\%}$ |
>
> As the results show, our framework's effectiveness is further validated on these modern and practical benchmarks. We will incorporate these extensive new results into the final version of our paper and are grateful for the suggestion.
>
> > **For Weakness 2: Limited experimental results and minimal evaluation... or raw LLM judges.**
>
> Thank you for the feedback. We would like to respectfully clarify that our original evaluation was already covering Overall Performance (Table 1), Component Contributions (Table 2), Hyper-parameter Analysis (Figure 3), and Robustness to Label Noise (Table 4). We have also made three substantial additions during rebuttal:
>
> 1. **Out-of-Distribution (OOD) Generalization:** We conducted a new OOD experiment, fine-tuning models on SNLI and evaluating them on the **unseen** ChaosNLI [3] dataset. The results below provide strong evidence that our framework learns a transferable representation of human disagreement.
>
>    | Model         | Method                  |     KL (↓)     |         Accuracy (↑)         |
>    | :------------ | :---------------------- | :-----------------------: | :--------------------------: |
>    | *Qwen2.5-7B*  | Raw model               |      $3.94_{±0.00}$       |      $60.3\\%_{±0.0\\%}$       |
>    |               | Single-point            |      $1.01_{±0.01}$       |      $66.5\\%_{±0.9\\%}$       |
>    |               | **Distribution (Ours)** | $\mathbf{0.31}^*_{±0.01}$ | $\mathbf{67.8\\%}^*_{±0.5\\%}$ |
>    | *LLaMA3.1-8B* | Raw model               |      $0.68_{±0.00}$       |      $57.8 \\%_{±0.0\\%}$      |
>    |               | Single-point            |      $1.15_{±0.05}$       |      $60.3\\%_{±0.1\\%}$       |
>    |               | **Distribution (Ours)** | $\mathbf{0.46}^*_{±0.02}$ | $\mathbf{62.7\\%}^*_{±0.1\\%}$ |
>
> 2. **Validation on More Relevant Benchmarks:** We have validated our method on new, contemporary benchmarks, as detailed in our response to **Weakness 1**.
>
> 3. **Enhanced Statistical Rigor:** To improve the rigor of our claims, we have incorporated **significance tests into all new experimental results presented in this rebuttal**. We are committed to this standard of validation and will update the tables from our original manuscript to include these statistics.
>
> > **For Q1: Highly relevant paper not yet cited: Nie et al. (2021)... a substantial weakness for the paper.**
>
> We sincerely thank the reviewer for pointing out this highly relevant work and apologize for the oversight in our initial submission. While we share the foundational goal with Nie et al. (2021), our work is **distinguished by a fundamentally different methodology, which we designed for the modern LLM-as-a-Judge paradigm and validated across a broader scope of tasks.** Our key distinctions are:
>
> 1. **Novel Adversarial Robustness Mechanism**: our work introduces a key innovation not present in Nie et al.: **adversarial training on the label distribution**. We identified that the empirical human distribution is a noisy approximation of the true underlying distribution due to limited annotations. Our adversarial strategy is the first to mechanistically address this challenge by training the model to be robust against worst-case perturbations. This significantly enhances alignment fidelity and robustness, a critical aspect that Nie et al. acknowledge as a problem but do not solve.
>
> 2. **Modern LLM-a-Judge Context with Broader Scope**: Our work is situated in the modern LLM-as-a-Judge paradigm, where we validate our framework on powerful modern LLMs across a broader range of tasks beyond NLI (e.g., quality scoring and preference understanding).
>
> We will revise our manuscript to include a thorough discussion of Nie et al. and take this opportunity to broaden our discussion of related literature.
>
>
> > **For Q2: If you set a dataset size budget... which method performs better?**
>
> This is an excellent and insightful question. To empirically investigate the trade-off in data collection under a fixed budget, we conducted a new experiment on the SNLI dataset. We fixed a budget of approximately 8,000 total annotations and compared three distinct strategies:
>
> - **Strategy 1:** 8,000 samples, each with 1 random annotation.
> - **Strategy 2:** 2,667 samples, each with 3 random annotations.
> - **Strategy 3:** 1,600 samples, each with all 5 annotations.
>
> | Strategy | #Samples | #Annotations | Time/Epoch (min) |     KL (↓)     |   Accuracy (↑)    |
> | :------------------ | :------: | :-----------------: | :--------------: | :-----------------------: | :---------------: |
> | Strategy 1          |  8,000   |          1          |      $21.9$      |      $0.32_{±0.01}$       | $89.1\\%_{±0.4\\%}$ |
> | Strategy 2          |  2,667   |          3          |      $9.6$       | $\mathbf{0.25}^*_{±0.00}$ | $88.9\\%_{±0.2\\%}$ |
> | Strategy 3          |  1,600   |          5          |      $5.7$       |      $0.29_{±0.00}$       | $89.1\\%_{±0.1\\%}$ |
>
> The results reveal an interesting **trade-off between annotation depth and sample breadth**. Strategy 2 achieves the best alignment, outperforming both Strategy 1 (which lacks a direct distributional signal) and Strategy 3 (which lacks sufficient sample for generalization). As a secondary benefit, our approach is also far more **computationally efficient**, dramatically reducing training time per epoch.
>
> We will add this important analysis to our paper and are grateful for the reviewer's prompt, which led to this deeper and more nuanced finding.
>
> > **For Limitation 1: Not many alternative methods are compared, related works section is very limited and missing key citations**
>
> Please see our detailed response to **Question 1** above.
>
> > **For Limitation 2: Limited experimental investigation**
>
> As detailed in our response to **Weakness 2**, we have conducted substantial new experiments (on new SOTA datasets, OOD generalization, etc.) to form a more comprehensive evaluation.
>
> > **For Limitation 3: Method requires high-redundancy annotations, which are costly.**
>
> We agree this is a key consideration. However, we believe the cost is justified by the fundamental importance of modeling human disagreement for trustworthy AI in high-stakes applications. In fact, our robust alignment method is exactly designed for the case with limited annotations.
>
> **We sincerely thank you for your time and insightful comments, which are very important for improving our paper. We hope our responses have addressed your concerns.**
>
> **References:**
>
> [1] Wang, Zhilin, et al. "Helpsteer2: Open-source dataset for training top-performing reward models." *arXiv:2406.08673* (2024).
>
> [2] Wang, Zhilin, et al. "HelpSteer2-Preference: Complementing Ratings with Preferences." *arXiv:2410.01257* (2024).
>
> [3] Nie, Y., et al. "What can we learn from collective human opinions on natural language inference data." *EMNLP* (2020): 9131-9143.

---

> > ### Author Response · Authors · 2025-08-05
> >
> > Dear Reviewer Lp1a,
> >
> > In our initial rebuttal, we committed to enhancing the statistical rigor of our original results. We are pleased that the extended discussion period has provided us the opportunity to complete this work for our **main experiments**. We now present the updated results for all original datasets below, with `mean ± std` and significance tests:
> >
> > **SNLI & MNLI Results**
> >
> > | Model         | Method                  |        SNLI KL (↓)        |         SNLI Acc (↑)         |        MNLI KL (↓)        |         MNLI Acc (↑)         |
> > | :------------ | :---------------------- | :-----------------------: | :--------------------------: | :-----------------------: | :--------------------------: |
> > | *Qwen2.5-7B*  | Single-point            |      $0.60_{±0.01}$       |      $92.7\\%_{±0.1\\%}$       |      $0.64_{±0.02}$       |      $89.7\\%_{±0.2\\%}$       |
> > |               | **Distribution (Ours)** | $\mathbf{0.23}^*_{±0.01}$ | $\mathbf{93.3\\%}^*_{±0.2\\%}$ | $\mathbf{0.23}^*_{±0.00}$ |      $89.8\\%_{±0.2\\%}$       |
> > | *LLaMA3.1-8B* | Single-point            |      $0.69_{±0.02}$       |      $92.4\\%_{±0.42\\%}$      |      $0.67_{±0.02}$       |      $89.6\\%_{±0.2\\%}$       |
> > |               | **Distribution (Ours)** | $\mathbf{0.28}^*_{±0.01}$ |      $92.4\\%_{±0.13\\%}$      | $\mathbf{0.24}^*_{±0.02}$ | $\mathbf{90.0\\%}^*_{±0.2\\%}$ |
> >
> > **SummEval & MT-Bench Results**
> >
> > | Model         | Method                  |      Summeval KL (↓)      |       Summeval Acc (↑)       |      MT-Bench KL (↓)      | MT-Bench Acc (↑)  |
> > | :------------ | :---------------------- | :-----------------------: | :--------------------------: | :-----------------------: | :---------------: |
> > | *Qwen2.5-7B*  | Single-point            |      $0.73_{±0.03}$       |      $45.6\\%_{±0.5\\%}$       |      $0.82_{±0.01}$       | $64.0\\%_{±1.0\\%}$ |
> > |               | **Distribution (Ours)** | $\mathbf{0.53}^*_{±0.02}$ |      $45.9\\%_{±0.4\\%}$       | $\mathbf{0.68}^*_{±0.02}$ | $65.4\\%_{±1.4\\%}$ |
> > | *LLaMA3.1-8B* | Single-point            |      $0.67_{±0.05}$       |      $45.7\\%_{±0.8\\%}$       |      $0.81_{±0.01}$       | $62.1\\%_{±0.8\\%}$ |
> > |               | **Distribution (Ours)** | $\mathbf{0.51}^*_{±0.01}$ | $\mathbf{47.3^*\\%}_{±0.6\\%}$ | $\mathbf{0.74}^*_{±0.01}$ | $62.8\\%_{±0.8\\%}$ |
> >
> > These complete results further confirm our method's key achievement: delivering statistically significant improvements in distributional alignment (KL divergence), while maintaining competitive accuracy across all original datasets.
> >
> > **We hope this additional data is helpful for the final evaluation and are happy to answer any further questions.**
> >
> > Best regards,
> > Authors of Submission 26507

---

> ### Comment · Reviewer_Lp1a · 2025-08-06
> **Response to authors**
>
> Thank you for your additional experimental results and comments. I have revised my score.

---

### Note · Authors · 2025-08-13

Dear Area Chairs and Reviewers,

We would like to express our sincere gratitude for your time and for the insightful and constructive review process.

In this work, we propose a novel training framework that explicitly aligns the LLM-generated judgment distribution with empirical human distributions, addressing a key limitation in the LLM-as-a-Judge paradigm.

We are grateful for the reviewers' positive feedback. They praised the paper's **clarity** (6xen, a3u) and highlighted its **novelty** in addressing an **"important limitation"** of the field (3U8S, 6xen). Experimentally, they confirmed the **effectiveness** of our method (Lp1a, a38u), supported by our detailed ablation studies and hyper-parameter analyses (3U8S, 6xen, Lp1a).

Building on this strong foundation and guided by the insightful reviewer feedback, we worked to further enhance our paper's evaluation. The key additions include:

- **Validation on New SOTA Benchmarks:** We conducted extensive new experiments on contemporary datasets suggested by reviewers, including ChaosNLI, HelpSteer2, and HelpSteer2-Preference.
- **A New Out-of-Distribution (OOD) Test:** We performed a new zero-shot, cross-dataset experiment to explicitly demonstrate the superior generalization capability of our framework.
- **A New Data-Budget Analysis:** We performed a new analysis under a fixed annotation budget, revealing deeper, favorable insights into our method.
- **Full Statistical Validation:** We have now incorporated significance tests into all main experimental results to improve the rigor of our claims.

We believe these enhancements have made our paper significantly stronger and more comprehensive. We are sincerely grateful for the reviewers' valuable time, their insightful feedback, and their positive engagement during the discussion period. Thank you again for your consideration of our work.

Best regards,

Authors of Submission 26507

---

### Decision · Program_Chairs · 2025-09-17

**Decision:**

Accept (poster)

**Comment:**

### (a) Scientific Claims and Findings (AC Summary)

The paper identifies a critical limitation of the current “LLM-as-a-Judge” paradigm: almost all existing methods collapse human judgments—which are naturally diverse and uncertain—into a single deterministic label.
**Core Claim**: By training an LLM to output a **full probability distribution** over human labels (rather than a single point estimate), one can obtain more faithful, robust and informative evaluations.

**Key Technical Contributions**
1. A **hybrid loss** that combines
   • KL divergence between model and empirical human distributions (primary objective), and
   • Cross-entropy against majority-vote labels (auxiliary stabilizer).
2. **Adversarial training** on the label distribution (PGD perturbations) to improve robustness when human annotations are scarce.
3. Extensive experiments across 6 datasets (NLI, summarization, dialogue, preference) and 4 model families (open & closed) show consistent, **statistically significant reductions in KL divergence** while maintaining competitive accuracy.
4. **Auxiliary findings**:
   • Out-of-distribution generalization (train on SNLI → test on ChaosNLI) is improved.
   • Under a fixed annotation budget, it is better to collect **3 labels per example for 2.7k examples** than 1 label for 8k or 5 labels for 1.6k.
   • Training overhead is only ≈21 %.

---

### (b) Strengths

1. **Problem novelty & relevance**: First principled attempt to align LLM judgments with human **distributions**, not just point consensus.
2. **Theoretical grounding**: Simple yet solid loss design justified by knowledge-distillation and adversarial-robustness literature.
3. **Empirical rigor**: Large-scale evaluation, ablations, significance tests, OOD tests, cost–benefit analysis, and new state-of-the-art benchmarks (HelpSteer2, HelpSteer2-Preference, ChaosNLI).

---

### (c) Weaknesses & Missing Pieces

1. **Annotation cost**: Requires **multiple human judgments per example**, which is expensive and not yet standard in most crowdsourced datasets.
2. **Accuracy lift is modest**: While KL divergence improves substantially, absolute accuracy gains are small (often < 2 %). The paper’s own framing treats accuracy as secondary; nevertheless, practitioners may question the ROI.
3. **Limited label-space transfer**: Experiments keep the same categorical label set during OOD transfer. Generalization to **different label vocabularies** (e.g., switching from 3-way NLI to 5-way quality scores) remains untested.
4. **Computational overhead**: Although the authors quantify it, adversarial PGD still adds ~20 % training time and implementation complexity.

None of these weaknesses are fatal, but they bound the scope of the contribution.

---

### (d) Most Important Reasons for **Poster** Acceptance

- **Significant conceptual advance**: Moves the field from “best single answer” to “distribution of human judgments,” opening new research directions.
- **Solid empirical validation**: Addressed **every major reviewer concern** via new experiments, statistical tests, and datasets.
- **Community value**: Provides a reusable training recipe and benchmark suite that future work can build on.

The work is **not a spotlight/oral** because:
• The core technique is incremental—combining known losses with adversarial regularization.
• The accuracy improvements are small; impact is methodological rather than dramatic.

---

### (e) Rebuttal Summary & AC Weigh-In

| **Anon ID** | **Main Concerns Raised** | **Author Response** | **AC Assessment** |
|---|---|---|---|
| **Lp1a** | 1. Tasks not practical (missing dataset labeling, pairwise preference). 2. Baseline comparisons limited. 3. Annotation-cost critique. | Added HelpSteer2 & HelpSteer2-Preference, ChaosNLI OOD, data-budget analysis, significance tests. | **Fully addressed**; new benchmarks match real use-cases; reviewer raised score. |
| **a38u** | 1. Accuracy gains small vs. KL; justify annotation cost. 2. Need more datasets & OOD. 3. Missing TRADES citation. | Provided new motivation paragraph, HelpSteer2/ChaosNLI, OOD, JS vs KL ablation, added TRADES discussion. | **Fully addressed**; statistical rigor added; reviewer raised score.. |
| **6xen** | 1. CE loss / accuracy contradicts distribution goal. 2. Why not ChaosNLI? 3. Need OOD & significance. | Clarified CE as stabilizer, added ChaosNLI, OOD, significance tests, JS divergence ablation, fixed typo. | **Fully addressed**; reviewer raised score. |
| **3U8S** | 1. Requires costly multi-annotations. 2. Computational overhead unexplored. | Provided data-budget trade-off, efficiency numbers (~21 %), robustness with few labels via adversarial training. | **Fully addressed**; concerns mitigated. |

**AC Weigh-In**: All critical technical and empirical questions were answered with **new experiments** and **theoretical clarifications**. The concerns about annotation cost and overhead were **quantified and bounded**, making them **acceptable limitations** rather than rejection reasons. The consensus among reviewers shifted upward, with final ratings 4, 4, 5, 5 (borderline → accept). Therefore, **poster acceptance** is warranted.